# MYC overrides HIF-1α to regulate proliferating primary cell metabolism in hypoxia

Courtney A Copeland[1,2], Benjamin A Olenchock[1,2], David Ziehr[1,2,3], Sarah McGarrity[1,2,4], Kevin Leahy[1,2], Jamey D Young[5], Joseph Loscalzo[1,2], William M Oldham[1,2]*

[1]Department of Medicine, Brigham and Women's Hospital, Boston, United States; [2]Department of Medicine, Harvard Medical School, Boston, United States; [3]Department of Medicine, Massachusetts General Hospital, Boston, United States; [4]Center for Systems Biology, School of Health Sciences, University of Iceland, Reykjavik, Iceland; [5]Departments of Chemical & Biomolecular Engineering and Molecular Physiology & Biophysics, Vanderbilt University, Nashville, United States

*For correspondence:
woldham@bwh.harvard.edu

Competing interest: The authors declare that no competing interests exist.

**Abstract** Hypoxia requires metabolic adaptations to sustain energetically demanding cellular activities. While the metabolic consequences of hypoxia have been studied extensively in cancer cell models, comparatively little is known about how primary cell metabolism responds to hypoxia. Thus, we developed metabolic flux models for human lung fibroblast and pulmonary artery smooth muscle cells proliferating in hypoxia. Unexpectedly, we found that hypoxia decreased glycolysis despite activation of hypoxia-inducible factor 1α (HIF-1α) and increased glycolytic enzyme expression. While HIF-1α activation in normoxia by prolyl hydroxylase (PHD) inhibition did increase glycolysis, hypoxia blocked this effect. Multi-omic profiling revealed distinct molecular responses to hypoxia and PHD inhibition, and suggested a critical role for MYC in modulating HIF-1α responses to hypoxia. Consistent with this hypothesis, MYC knockdown in hypoxia increased glycolysis and MYC over-expression in normoxia decreased glycolysis stimulated by PHD inhibition. These data suggest that MYC signaling in hypoxia uncouples an increase in HIF-dependent glycolytic gene transcription from glycolytic flux.

## Editor's evaluation

The manuscript by Copeland and colleagues describes the impact of HIF1a, MYC and metabolism in pulmonary lung fibroblast and pulmonary artery smooth muscle cell phenotype, which is highly relevant to pulmonary vascular disease. The work includes metabolic flux assays of cultured cells, using a combination of metabolite concentration assessments, stable isotope-labeled substrates coupled with mass spectrometry, mathematical modeling, and cell proliferation analysis. Overall the findings are that there is an unexpected drop in lactate production in hypoxia and with HIF augmentation. These studies will add to the field's understanding of the role of HIF and cellular metabolism in pulmonary hypertension.

## Introduction

Cellular responses to hypoxia propel many physiologic and pathologic processes from wound healing and angiogenesis to vascular remodeling and fibrosis (*Lee et al., 2019b*; *Semenza, 2012*). These activities require cells to continue energetically demanding tasks, such as macromolecular biosynthesis

and proliferation, despite limited oxygen availability. Since respiration is the most efficient way for cells to produce energy, cell metabolism must adapt to meet energy demands when oxygen supply is limiting. Understanding how these metabolic adaptations sustain critical cellular processes in hypoxia is fundamentally important to our understanding of human health and disease.

Cells typically respond to hypoxia by shifting energy production away from respiration and toward glycolysis. This response is mediated primarily by stabilization of the hypoxia-inducible transcription factor 1α (HIF-1α). HIF-1α activates the transcription of glucose transporters, glycolytic enzymes, and lactate dehydrogenase, while decreasing the expression of tricarboxylic acid (TCA) cycle and electron transport chain enzymes (*Lee et al., 2020*; *Semenza, 2012*). Although HIF-1α is constitutively expressed, it is hydroxylated by prolyl hydroxylase enzymes (PHDs) in normoxia and targeted for proteasomal degradation. PHDs are α-ketoglutarate-dependent dioxygenase enzymes that require molecular oxygen for their enzymatic activity. When oxygen tension falls, PHD activity decreases, leading to HIF-1α stabilization and activation of its downstream transcriptional program. Overall, this transcriptional program should increase glycolytic flux and lactate production while decreasing TCA cycle flux and oxidative phosphorylation.

In addition to metabolic changes designed to maintain energy supply, hypoxic cells also reduce energy demand through down-regulation of $Na^+/K^+$-ATPase, slowing protein translation, and decreasing cell proliferation (*Hubbi and Semenza, 2015*; *Wheaton and Chandel, 2011*). In particular, HIF-1α decreases cell proliferation by activating cyclin-dependent kinase inhibitor expression, inhibiting cell-cycle checkpoint progression (*Gardner et al., 2001*), and antagonizing pro-proliferative MYC signaling (*Koshiji et al., 2004*). Despite these canonical effects of HIF-1α activation, there are many examples where cells continue to proliferate despite hypoxic stress, including cancer cells, stem cells, and lung vascular cells (*Hubbi and Semenza, 2015*). How these cells meet the metabolic needs of sustained proliferation in hypoxia is an active area of investigation (*Jain et al., 2020*; *Lee et al., 2020*; *Oldham et al., 2015*). Since hypoxia is a prominent feature of cancer biology as tumor growth outstrips blood supply, most detailed metabolic studies of hypoxic cell metabolism have used tumor cell models, yielding important insights into the metabolic pathobiology of cancer (*Garcia-Bermudez et al., 2018*; *Jiang et al., 2016*; *Lee et al., 2019a*; *Meléndez-Rodríguez et al., 2019*; *Metallo et al., 2011*; *Wise et al., 2011*). For example, stable isotope tracing and metabolic flux analyses identified a critical role for the reductive carboxylation of glutamine-derived α-ketoglutarate for lipid biosynthesis in supporting tumor growth (*Gameiro et al., 2013*; *Metallo et al., 2011*; *Scott et al., 2011*; *Wise et al., 2011*), and metabolomic studies identified aspartate as a limiting metabolite for cancer cell proliferation under hypoxia (*Garcia-Bermudez et al., 2018*). By contrast, comparatively little is known about metabolic adaptations of primary cells to hypoxia. Indeed, the importance of reductive carboxylation or aspartate biosynthesis remains to be elucidated in primary cells. A more complete understanding of primary cell metabolic adaptations to hypoxia would provide an important context for understanding how metabolic reprogramming supports normal cellular responses to hypoxia, how these responses may be (mal)adaptive in a variety of disease contexts, and how the hypoxia metabolic program in primary cells differs from that observed in cancer cells.

To address these questions, we have developed models of bioenergetic carbon flux in human lung fibroblasts (LFs) and pulmonary artery smooth muscle cells (PASMCs) cultured in 21% or 0.5% oxygen. These cells may be exposed to a wide range of oxygen concentrations in vivo, continue to proliferate despite hypoxic culture conditions in vitro, and play important roles in the pathology of non-cancerous diseases in which tissue hypoxia features prominently, including pulmonary hypertension and pulmonary fibrosis. We found that hypoxia fails to increase glycolysis in these primary cells despite robust up-regulation of the HIF-1α transcriptional program. In normoxia, HIF-1α stabilization by the PHD inhibitor molidustat (BAY-85–3934, "BAY") (*Flamme et al., 2014*) did increase glycolysis and lactate efflux; however, hypoxia blocked this response. These findings suggested that important hypoxia-dependent regulatory mechanisms override the metabolic consequences of HIF-1α-dependent increases in glycolytic gene expression. Transcriptomic profiling identified a critical role for the transcription factor MYC in the adaptive response to hypoxia. Using knockdown and overexpression approaches, we demonstrated that MYC attenuates HIF-driven glycolysis in hypoxia and following HIF stabilization in normoxia.

## Results

### Hypoxia uncouples HIF-dependent glycolytic gene expression from glycolytic metabolic flux

The goal of this study was to characterize hypoxia-induced metabolic changes in proliferating primary LFs and PASMCs. To accomplish this goal, we used metabolic flux analysis to model how cell metabolism supports cell proliferation. Metabolic flux analysis fits cell proliferation rate, extracellular flux measurements, and $^{13}C$ isotope labeling patterns to a computational model of cell metabolism (*Antoniewicz, 2018*). This analysis reconstructs comprehensive flux maps that depict the flow of carbon from extracellular substrates through intracellular metabolic pathways into cell biomass and metabolic by-products (*Young, 2014*). These models assume that cells are at a metabolic pseudo-steady state over the experimental time course (*Buescher et al., 2015*). Exponential growth phase is thought to reflect metabolic pseudo-steady state as cells in culture steadily divide at their maximal condition-specific rate, provided nutrient supply does not become limiting (*Ahn and Antoniewicz, 2011*; *Buescher et al., 2015*). Thus, we first set out to define experimental conditions to capture exponential growth phase in normoxic and hypoxic cultures.

Cells were seeded and placed into hypoxia for 24 hr prior to sample collection to provide adequate time for activation of the hypoxia-dependent transcriptional program (*Figure 1A*). We selected 0.5% oxygen for hypoxia as this level yielded the most reproducible phenotypic differences compared to 21% oxygen, while being physiologically relevant and above the $K_M$ of cytochrome *c* oxidase (electron transport chain complex IV) for oxygen (*Lee et al., 2020*; *Wenger et al., 2015*). We identified the optimal cell seeding density and time course for exponential cell growth (*Figure 1B*). Hypoxia decreased cell proliferation rates (*Figure 1C*), but slower growth was not associated with decreased cell viability (*Figure 1—figure supplement 1A*). As anticipated, hypoxic cells demonstrated robust stabilization of HIF-1α associated with up-regulation of its downstream targets glucose transporter 1 (GLUT1) and lactate dehydrogenase A (LDHA; *Figure 1D–H*). These changes persisted for the duration of the experimental time course.

We next determined the extracellular fluxes of glucose (GLC), lactate (LAC), pyruvate (PYR), and amino acids (*Figure 1I–J*). Flux calculations incorporated cell growth rate, extracellular metabolite concentrations, metabolite degradation rates, and medium evaporation rate (see Materials and methods; *Murphy and Young, 2013*; *Figure 1—figure supplement 1B–C*). Interestingly, while we observed a modest increase in glucose uptake, we found that hypoxia actually decreased lactate efflux (*Figure 1I*). This finding was confirmed by measuring the rate of [U-$^{13}C_3$]-lactate produced from LFs cultured with [U-$^{13}C_6$]-glucose (*Figure 1—figure supplement 2*). Hypoxia decreased lactate efflux despite activating HIF-1α and increasing glycolytic enzymes expression (*Figure 1D–H*).

To test if more severe hypoxia would augment glycolysis, we cultured cells in 0.2% ambient oxygen (*Figure 1—figure supplement 3*). Under these conditions, we observed no change in glucose or lactate fluxes, similar to 0.5% oxygen culture. To test if this unexpected response was unique to LFs, we studied PASMCs under 0.5% oxygen conditions (*Figure 1—figure supplement 4*). Similar to LFs, we observed no change in glucose uptake and reduced lactate efflux in PASMCs. Together, these data suggest that hypoxia uncouples HIF-1α target gene expression and glycolytic flux in proliferating primary cells.

Since hypoxia did not increase glycolysis in LFs, we wanted to determine how these cells responded to HIF-1α stabilization in normoxia. To activate HIF-1α, LFs were treated with the PHD inhibitor molidustat (BAY, 10 µM) using a similar time course as our hypoxia experiments (*Figure 2*). Like hypoxia, BAY decreased cell growth rate (*Figure 2B–C*) and activated the HIF-1α transcriptional program (*Figure 2D–H*). Unlike hypoxia, HIF-1α stabilization in normoxia markedly increased glucose uptake and lactate efflux (*Figure 2I*). Although hypoxia and BAY treatments both increased HIF-1α, GLUT1, and LDHA to a similar degree, the glycolytic metabolic response differed markedly between these treatments.

### Extracellular fluxes are treatment and cell-type dependent

In addition to glucose and lactate, we also determined the extracellular fluxes of pyruvate and amino acids (*Figure 1J*, *Figure 1—figure supplement 3J*, *Figure 1—figure supplement 4J*, *Figure 2J*). To our knowledge, this is the first comprehensive extracellular flux profiling of key metabolic substrates in primary cells. In LFs, changes in extracellular fluxes were modest overall, with hypoxia generally

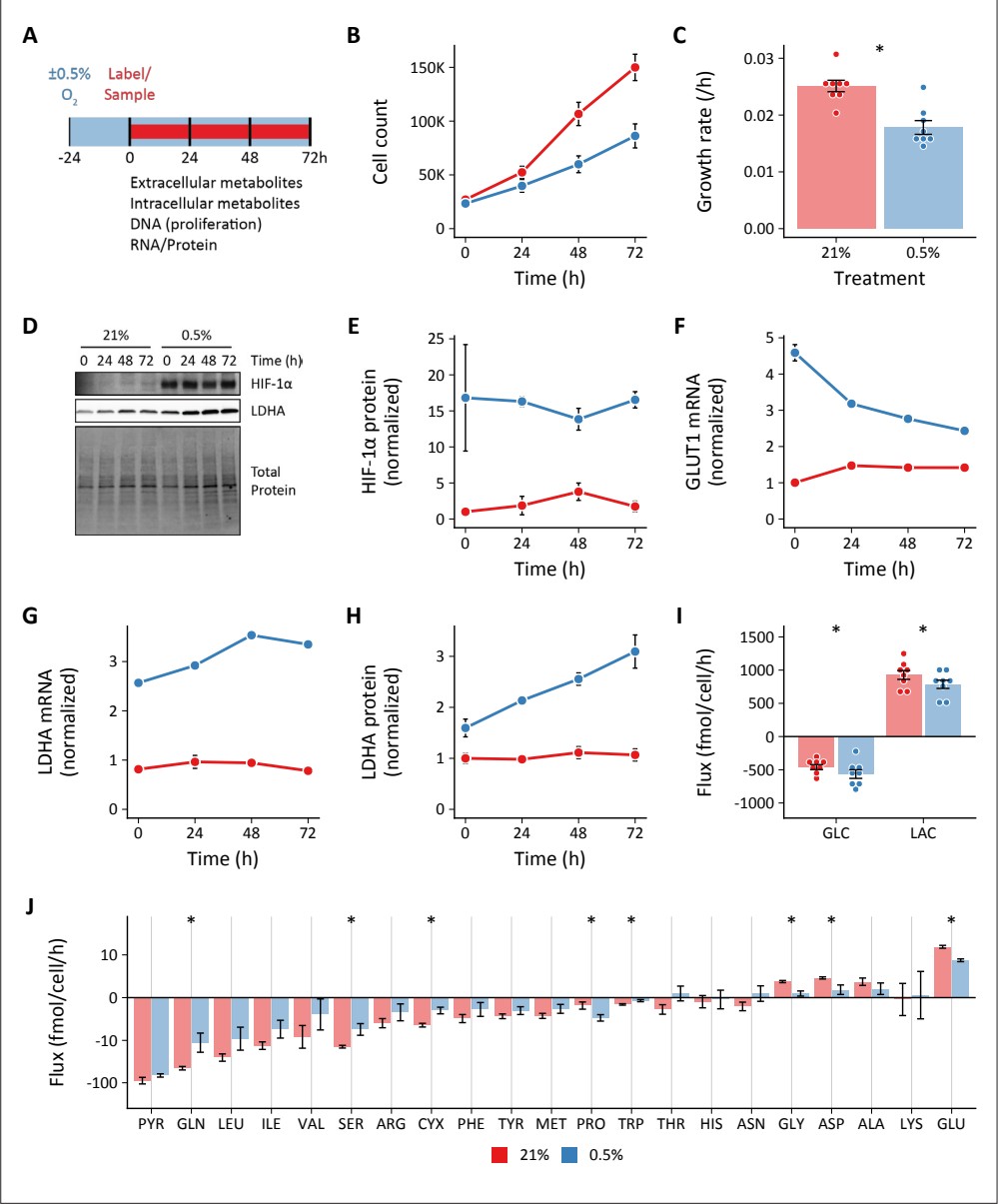

**Figure 1.** Effects of 0.5% oxygen on extracellular metabolite fluxes in lung fibroblasts. (**A**) Lung fibroblasts (LFs) were cultured in 21% or 0.5% oxygen beginning 24 hr prior to time 0. Samples were collected every 24 hr for 72 hr. (**B**) Growth curves of LFs in each experimental condition (n=8). (**C**) Growth rates from (**B**) were determined by robust linear modeling of log-transformed growth curves. (**D**) Representative immunoblot of LF protein lysates cultured as in (**A**). (**E**) Relative change in HIF-1α protein levels from (**D**) normalized to 21% oxygen at time 0 (n=4). (**F**) Relative change in GLUT1 mRNA levels normalized to 21% oxygen treatment at time 0 (n=4). (**G**) Relative change in LDHA mRNA levels as in (**F**). (**H**) Relative change in LDHA protein levels as in (**E**). (**I**) Extracellular fluxes of glucose (GLC) and lactate (LAC) (n=8). By convention, negative fluxes indicate metabolite consumption. (**J**) Extracellular fluxes of pyruvate (PYR) and amino acids. Data are mean ± SEM (* p<0.05).

The online version of this article includes the following source data and figure supplement(s) for figure 1:

**Source data 1.** Uncropped blot images for *Figure 1*.

**Figure supplement 1.** Supporting data for extracellular flux calculations.

**Figure supplement 2.** Quantifying lactate efflux generated by [U-¹³C₆]-glucose.

**Figure supplement 3.** Effects of 0.2% oxygen on extracellular metabolite fluxes in lung fibroblasts.

**Figure supplement 3—source data 1.** Uncropped blot images for *Figure 1—figure supplement 3*.

*Figure 1 continued on next page*

*Figure 1 continued*

**Figure supplement 4.** Effects of 0.5% oxygen on extracellular metabolite fluxes in pulmonary artery smooth muscle cells.

**Figure supplement 4—source data 1.** Uncropped blot images for *Figure 1—figure supplement 4*.

decreasing the fluxes of all measured metabolites. These findings were similar with 0.2% oxygen (*Figure 1—figure supplement 3J*).

Notably, we observed a significant decrease in glutamine consumption in hypoxic LFs. This finding contrasts with previous studies of cancer cell metabolism demonstrating increased glutamine uptake as a key feature of the metabolic response to hypoxia (*Gameiro et al., 2013*; *Metallo et al., 2011*; *Wise et al., 2011*). In these systems, glutamine-derived α-ketoglutarate was reductively carboxylated by isocitrate dehydrogenase enzymes to generate citrate for lipid biosynthesis. Glutamine has also been shown to support TCA cycling in hypoxia in a Burkitt lymphoma model (*Le et al., 2012*). Unlike LFs, PASMCs did exhibit a trend toward increased glutamine uptake (*Figure 1—figure supplement 4J*). To examine the relative importance of glucose and glutamine to the proliferation of these cells in hypoxia, we measured LF and PASMC growth rates in the absence of either substrate (*Figure 3*). In LFs, absence of either glucose or glutamine reduced cell proliferation to a similar extent (*Figure 3A*). In hypoxia, glucose deficiency decreased LF proliferation rate further, while glutamine deficiency had no additional impact. These findings are consistent with extracellular flux measurements demonstrating decreased glutamine consumption by LFs in hypoxia. Interestingly, neither glucose nor glutamine deficiency decreased PASMC proliferation (*Figure 3B*), suggesting a high degree of metabolic flexibility in these cells.

In LFs, among all of the measured amino acid fluxes, proline consumption uniquely increased (*Figure 1J*). Hypoxia increases collagen expression in these cells (*Liu et al., 2013*) and proline constitutes ~10% of the total amino acid content of collagens. Together, these data suggest an important contribution of extracellular proline to collagen production in hypoxic LFs as has been observed in other fibroblast cell lineages (*Szoka et al., 2017*).

In PASMCs, we observed increased consumption of the branched-chain amino acids (BCAAs) leucine and valine as well as arginine (*Figure 1—figure supplement 4J*), which was not observed in LFs. BCAAs are transaminated by branch chain amino transferase enzymes to branched chain α-keto acids (BCKAs). BCKAs are further metabolized to yield acyl-CoA derivatives for lipogenesis or oxidation (*Crown et al., 2015*; *Mann et al., 2021*). Previous studies have shown that hypoxia up-regulates arginase expression in hypoxic PASMCs (*Chen et al., 2009*; *Xue et al., 2017*) to support polyamine and proline synthesis required for cell proliferation (*Li et al., 2001*). Interestingly, activation of these metabolic pathways in hypoxia was not observed in LFs and suggests distinct metabolic dependencies of these different cell types.

Compared to hypoxia treatment, BAY demonstrated more modest effects on amino acid fluxes generally (*Figure 2J*). In particular, glutamate efflux was not affected by BAY treatment, while it was reduced by hypoxia. Alanine efflux was increased by BAY treatment, but decreased by hypoxia. In addition to the glucose and lactate fluxes noted above, these findings further highlight fundamental differences in the metabolic consequences of HIF-1α activation in normoxia and hypoxia.

## Isotope tracing reveals altered substrate utilization in hypoxia

To investigate intracellular metabolic reprogramming in hypoxic cells, we performed $^{13}$C stable isotope tracing with [U-$^{13}$C$_6$]-glucose, [1,2-$^{13}$C$_2$]-glucose, and [U-$^{13}$C$_5$]-glutamine. Isotopic enrichment of downstream metabolites in glycolysis and the TCA cycle were determined by LC-MS (*Figure 4—figure supplement 1*, *Figure 4—figure supplement 2*). Small changes in the patterns of isotope incorporation were observed following hypoxia or BAY treatment. The most substantial differences were observed in pyruvate (PYR), the terminal product of glycolysis, and citrate (CIT), a central metabolic node in TCA and fatty acid metabolism (*Figure 4A–C*). Both hypoxia and BAY treatments decreased incorporation of glucose-derived carbon into pyruvate (*Figure 4A*; i.e. the unlabeled, or M0, fraction was greater). This suggests an increased contribution from an unlabeled carbon source, such as extracellular pyruvate, lactate, or alanine to the intracellular pyruvate pool following PHD inhibition.

Total citrate labeling from [U-$^{13}$C$_6$]-glucose was unchanged across the treatment conditions (*Figure 4B*). We observed decreased M2 and M4 citrate isotopes, consistent with decreased pyruvate

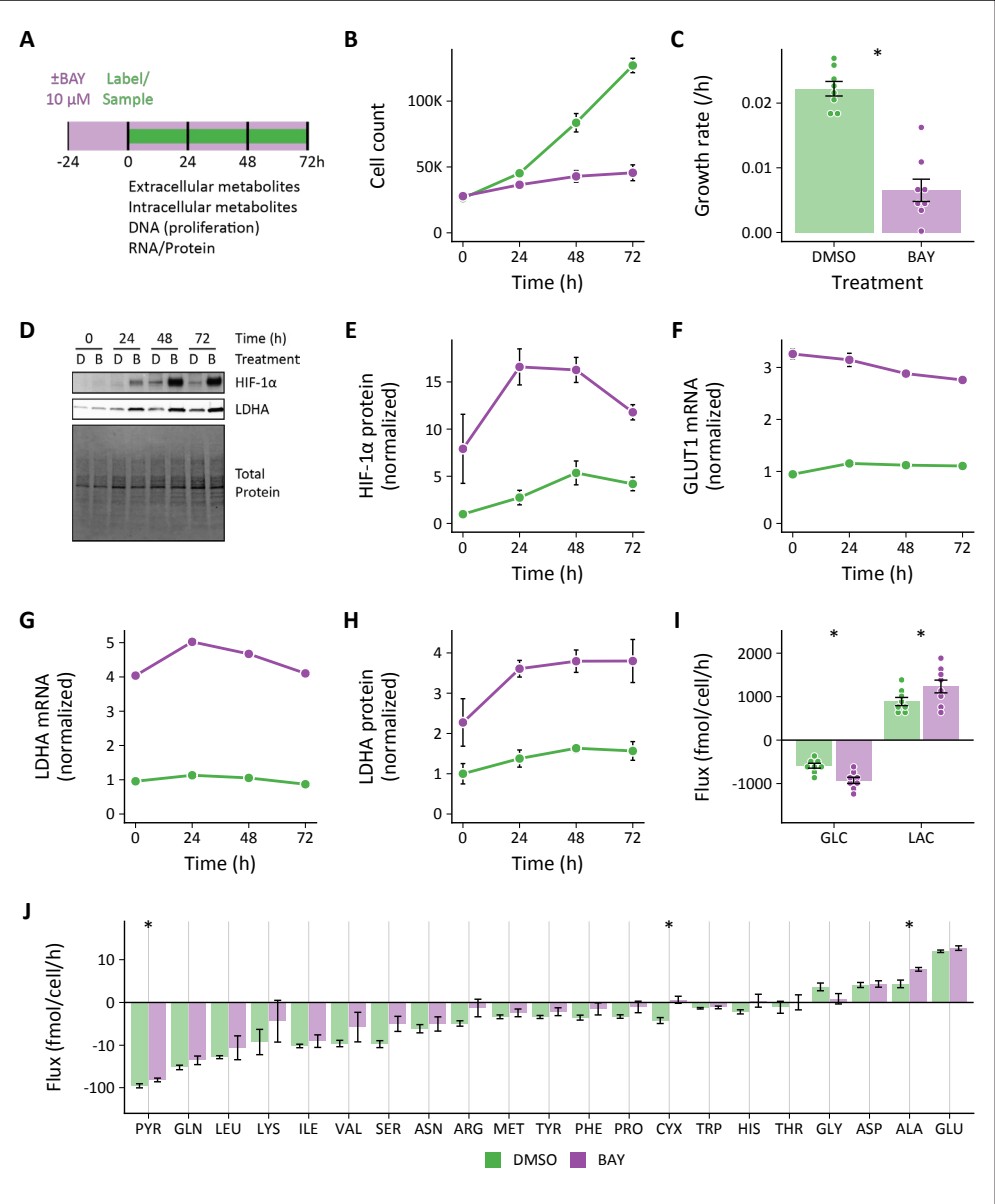

**Figure 2.** Effects of pharmacologic prolyl hydroxylase inhibition on extracellular metabolite fluxes in lung fibroblasts. (**A**) Lung fibroblasts (LFs) were treated with the prolyl hydroxlyase inhibitor molidustat (BAY, 10 µM) or DMSO beginning 24 hr prior to time 0. Samples were collected every 24 hr for 72 hr. (**B**) Growth curves of LFs in each experimental condition (n=8). (**C**) Growth rates from (**B**). (**D**) Representative immunoblot of LF protein lysates cultured as in (**A**). (**E**) Relative change in HIF-1α protein levels from (**D**) normalized to DMSO at time 0 (n=4). (**F**) Relative change in GLUT1 mRNA levels normalized to DMSO at time 0 (n=4). (**G**) Relative change in LDHA mRNA levels as in (**F**). (**H**) Relative change in LDHA protein levels as in (**E**). (**I**) Extracellular fluxes of glucose (GLC) and lactate (LAC) (n=8). By convention, negative fluxes indicate metabolite consumption. (**J**) Extracellular fluxes of pyruvate (PYR) and amino acids. Data are mean ± SEM (* p<0.05).

The online version of this article includes the following source data for figure 2:

**Source data 1.** Uncropped blot images for *Figure 2*.

dehydrogenase activity in hypoxia. Interestingly, we observed increased M3 and M5 citrate isotopes. Pyruvate carboxylase catalyzes the carboxylation of pyruvate to oxaloacetate after which all three pyruvate carbons are incorporated into citrate by citrate synthase. Thus, this labeling pattern suggests a more prominent contribution of pyruvate carboxylase to sustain TCA cycle anaplerosis despite pyruvate dehydrogenase inhibition following HIF-1α activation. Compared to glucose, glutamine labeled

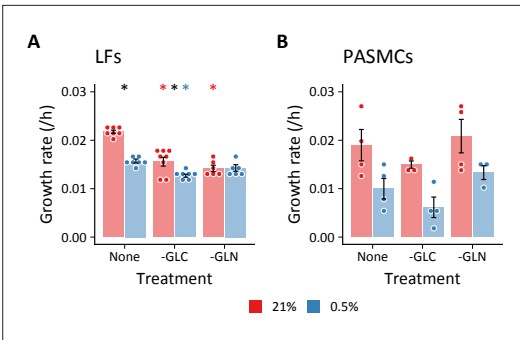

**Figure 3.** Cell growth rates following substrate deprivation. (**A, B**) Lung fibroblasts (n=8 biological replicates) (**A**) and pulmonary artery smooth muscle cells (n=4 biological replicates) (**B**) were cultured in MCDB131 medium lacking either glucose (-GLC) or glutamine (-GLN) for 72 hr. Growth rates were calculated from total DNA quantification. Data are mean ± SEM (* p<0.05; black compares 21% and 0.5% oxygen within a given treatment, red and blue compare substrate deficiency to replete medium in 21% and 0.5% oxygen, respectively.).

a smaller fraction of citrate, and this labeling was decreased substantially by hypoxia or BAY treatment (*Figure 4C*), suggesting a less important contribution of glutamine to TCA anaplerosis under these conditions. In addition, the overall fraction of M5 citrate resulting from reductive carboxylation of glutamine-derived α-ketoglutarate was low (<7%) (*Figure 4D*). Although a hypoxia-mediated increase in M5 citrate was observed, the overall fraction was much less than the 10–20% levels previously reported in cancer cells (*Metallo et al., 2011*; *Wise et al., 2011*).

The stable isotope labeling patterns in PASMCs were generally similar to LFs (*Figure 4—figure supplement 2*). The most notable differences between LF and PASMC labeling were observed in citrate. Compared with LFs, a much lower fraction of total citrate was labeled by glucose in PASMCs. Less activity of pyruvate carboxylase in these cells was suggested by decreased M3 and M5 citrate isotopes after glucose labeling. Interestingly, the M5 citrate fraction in PASMCs was more consistent with previous reports from the cancer literature (*Figure 4D*), suggesting activation of glutamine anaplerosis for biomass synthesis in these cells.

## Glycolytic flux in hypoxia is closely coupled to cell growth rate

The mass isotopomer distribution for a given metabolite is determined by the complex relationship among the rate of isotope incorporation into the metabolic network, the contributions of unlabeled substrates, and fluxes through related pathways. To clarify how these labeling patterns reflect changes in intracellular metabolite fluxes, we next generated metabolic flux models incorporating the extracellular flux measurements and stable isotope tracing data described above. Preliminary labeling time courses indicated that, even after 72 hr of labeling, intracellular metabolites did not reach isotopic steady state (*Figure 5—figure supplement 1*). Thus, we performed isotopically non-stationary metabolic flux analysis as implemented by Isotopomer Network Compartment Analysis (INCA; *Jazmin and Young, 2013*; *Murphy and Young, 2013*; *Young et al., 2014*).

Overall, LF and PASMC metabolic fluxes were dominated by high rates of glucose uptake and glycolysis (*Figure 5—figure supplement 2*). In normoxia, approximately 10% of cytoplasmic pyruvate enters the TCA cycle with the balance converted to lactate. Consistent with the extracellular flux measurements and isotope labeling patterns described above, hypoxia significantly decreased glycolysis, the TCA cycle, and amino acid metabolism (*Figure 5A*). A significant increase in pentose phosphate pathway flux was also observed, although the absolute flux through this pathway is low. By contrast, HIF-1α activation by BAY in 21% oxygen increased glycolysis and lactate fermentation by nearly 50% (*Figure 5B*), but had a similar effect as hypoxia on decreasing serine and glutamine uptake. Metabolite fluxes in DMSO-treated cells were similar to 21% oxygen controls (*Table 1*, *Table 2*).

In normoxia, the magnitude of intracellular metabolite fluxes was similar in LFs and PASMCs (*Figure 5—figure supplement 2*, *Table 1*, *Table 3*). Compared to LFs, PASMCs had slower rates of glycolysis and faster rates of TCA metabolism driven, in part, by increased glutamine uptake (*Figure 5—figure supplement 3*). In hypoxia, PASMCs exhibited similar decreases in glycolytic flux as LFs but also a marked, and unexpected, increase in TCA flux (*Figure 5—figure supplement 4*). The increased TCA flux in PASMCs was driven by increased glutamine consumption. This finding is similar to a prior report of glutamine-driven oxidative phosphorylation in hypoxic cancer cells (*Fan et al., 2013*), where oxidative phosphorylation continued to provide the majority of cellular ATP even at 1% oxygen.

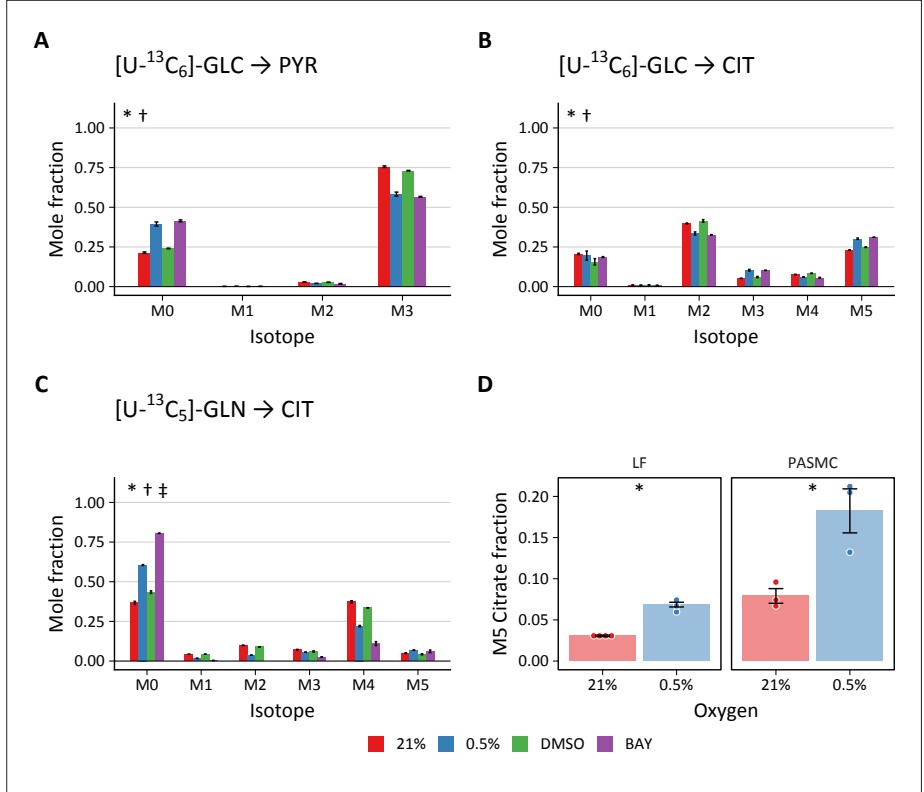

**Figure 4.** Stable isotope labeling of lung fibroblasts following hypoxic and pharmacologic PHD inhibition.
(**A**) Mass isotopomer distribution (MID) of pyruvate (PYR) following 72 hr labeling with [U-$^{13}C_6$]-glucose (GLC).
(**B**) MID of citrate after 72 hr labeling with [U-$^{13}C_6$]-GLC (**C**) MID of citrate after 72 hr labeling with [U-$^{13}C_5$]-glutamine
(GLN). Data are mean ± SEM (n=4, p<0.05 indicated as * 0.5% v. 21% oxygen, † BAY v. DMSO, ‡ Δoxygen v. ΔBAY).
(**D**) Fraction of M5 citrate indicating reductive carboxylation after labeling with [U-$^{13}C_5$]-GLN in LFs and PASMCs
(n=3–4, * p<0.05).

The online version of this article includes the following figure supplement(s) for figure 4:

**Figure supplement 1.** Mass isotopomer distributions after 72 hr of labeling in lung fibroblasts.

**Figure supplement 2.** Mass isotopomer distributions after 36 hr of labeling in pulmonary artery smooth muscle
cells.

Given the global decrease in bioenergetic metabolic flux in hypoxic LFs, we hypothesized that these differences may be a consequence of decreased growth rate. After normalizing metabolite fluxes in normoxia and hypoxia to the cell growth rate, a modest increase (~10%) in glycolytic flux was observed (*Figure 5—figure supplement 5*). This finding suggests that, while glycolysis increases relative to growth rate in hypoxic cells, regulators of cell proliferation rate override the anti-proliferative effects of the HIF-1α transcriptional program. Indeed, even after adjusting for cell growth rate, the relative increase in glycolytic flux is modest compared to the marked up-regulation of glycolytic proteins and the glycolytic potential of these cells demonstrated by BAY treatment in normoxia. BAY treatment decreased cell proliferation rate (*Figure 2B–C*), indicating that, unlike hypoxia, PHD inhibition in normoxia uncouples cell proliferation and metabolic flux.

## Hypoxia and BAY treatment increase lactate oxidation

One important feature of metabolic flux analysis is its ability to determine the individual forward and backward fluxes of bidirectional reactions, or the so-called exchange fluxes. Although the metabolite exchange fluxes for bidirectional reactions tend to be poorly resolved by this method (*Wiechert, 2007*), two observations from our models are noteworthy (*Table 1*, *Table 2*, *Table 3*). First, consistent with the stable isotope tracing results, the modeled rate of reductive carboxylation through reverse flux by isocitrate dehydrogenase in LFs is low (~4 fmol/cell/h), unchanged by hypoxia, and

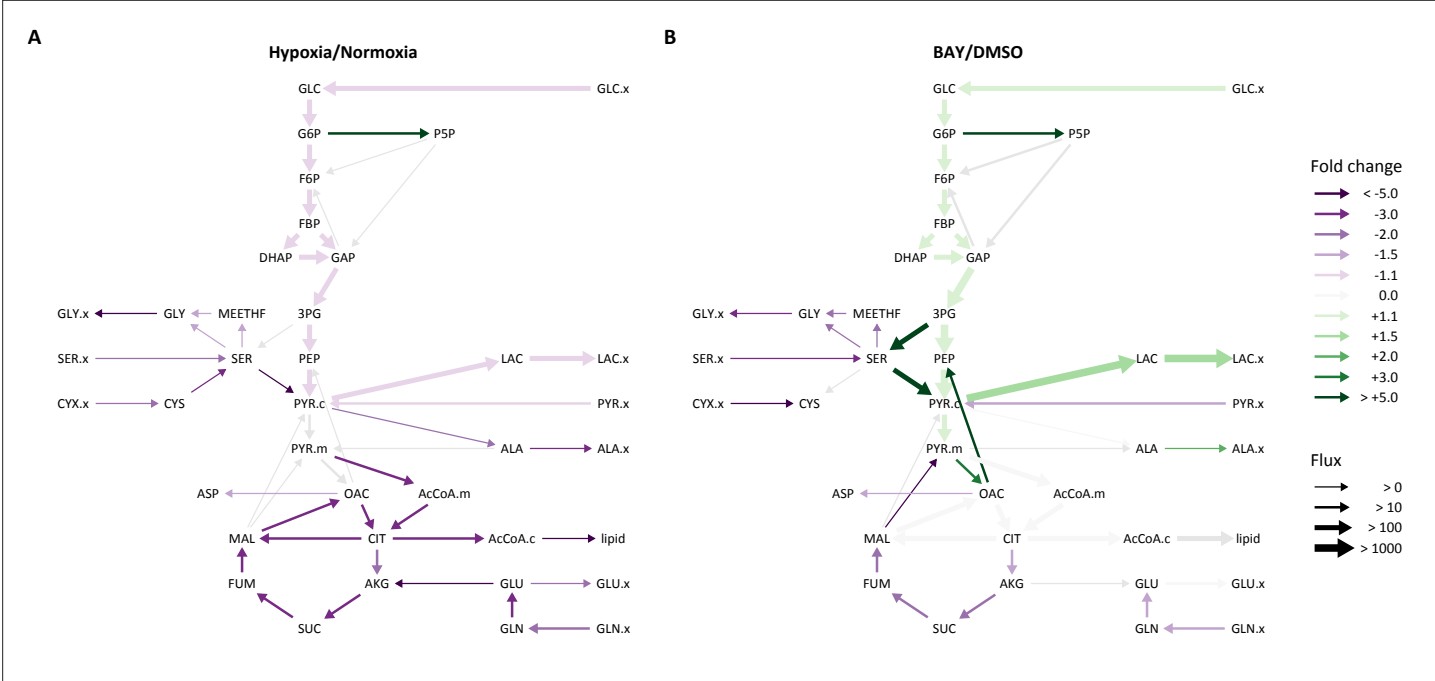

**Figure 5.** Metabolic flux analysis of lung fibroblasts following hypoxic and pharmacologic PHD inhibition. (**A**) Ratio of modeled metabolic fluxes in 0.5% oxygen compared to 21% oxygen. Fluxes with non-overlapping confidence intervals are highlighted with arrows colored according to the magnitude of the fold change. Arrow thickness corresponds to the absolute flux measured in hypoxia. (**B**) Ratio of metabolic fluxes in BAY-treated cells compared to DMSO-treated control. Arrows are colored as in (**A**) and arrow weights correspond to the absolute flux as measured in BAY-treated cells.

The online version of this article includes the following figure supplement(s) for figure 5:

**Figure supplement 1.** Isotope incorporation over the labeling time course.

**Figure supplement 2.** Isotopically non-stationary metabolic flux analysis of cell metabolism in 21% oxygen.

**Figure supplement 3.** Comparison of lung fibroblast and pulmonary artery smooth muscle cell metabolic fluxes.

**Figure supplement 4.** Comparison of pulmonary artery smooth muscle cell metabolic fluxes in 21% and 0.5% oxygen.

**Figure supplement 5.** Growth rate adjusted changes in hypoxic lung fibroblast metabolism.

modestly increased by BAY treatment. By contrast, the rate of reductive carboxylation increases sixfold in PASMCs in hypoxia, highlighting a potentially important role for this pathway in the metabolic response of PASMCs to decreased oxygen availability (*Figure 6*).

Second, PHD inhibition increases lactate transport exchange flux in LFs from ~0 to 1500 and 700 fmol/cell/h with 0.5% oxygen and BAY treatment, respectively, with similar results in PASMCs (*Figure 7A*). This observation suggests increased lactate uptake with hypoxia or BAY treatment. To investigate this hypothesis, LFs and PASMCs were treated with [U-$^{13}$C$_3$]-lactate (2 mM) and $^{13}$C incorporation into intracellular metabolites was analyzed by LC-MS (*Figure 7B*, *Figure 4—figure supplement 1*, *Figure 4—figure supplement 2*). Lactate labeled ~50% of citrate and ~20% of downstream TCA cycle metabolites (α-ketoglutarate, malate, aspartate) in both LFs and PASMCs, indicating that lactate may be an important respiratory fuel source in these cells even though lactate efflux is high. Although increased labeling of pyruvate was observed in hypoxic PASMCs, this increase did not flow through to TCA metabolites as observed in LFs (*Figure 4—figure supplement 2*). Lactate has been used less commonly than glucose and glutamine in stable isotope tracing studies. *Faubert et al., 2017* demonstrated lactate incorporation in human lung adenocarcinoma in vivo. In this study, lactate incorporation corresponded to regions of high glucose uptake as determined by [$^{18}$F]-fluorodeoxyglucose positron emission tomography, suggesting that lactate consumption can occur even in areas of high glucose utilization. Subsequently, investigators have demonstrated the importance of lactate as a metabolic fuel in vivo (*Hui et al., 2020*; *Hui et al., 2017*).

In addition to downstream metabolites, we also observed hypoxia- and BAY-dependent increases in lactate incorporation in fructose bisphosphate (FBP) and 3-phosphoglycerate (3PG). This observation is

**Table 1.** LF fluxes in 21% and 0.5% oxygen.

| Type | Pathway | ID | Reaction | 21% * | | | 0.5%† | | | Ratio |
|---|---|---|---|---|---|---|---|---|---|---|
| | | | | Flux | LB | UB | Flux | LB | UB | |
| NET | Transport | GLUT | GLC.x → GLC | 5.14E+02 | 5.11E+02 | 5.21E+02 | 4.41E+02 | 4.26E+02 | 4.58E+02 | 0.86 |
| | | PYRR | PYR.x → PYR.c | 7.56E+01 | 7.31E+01 | 7.96E+01 | 6.21E+01 | 5.83E+01 | 6.60E+01 | 0.82 |
| | | MCT | LAC ⇌ LAC.x | 9.99E+02 | 9.98E+02 | 1.02E+03 | 8.91E+02 | 8.62E+02 | 9.25E+02 | 0.89 |
| | | ALAR | ALA → ALA.x | 2.25E+00 | 1.95E+00 | 2.49E+00 | 5.84E-01 | 1.10E-03 | 1.16E+00 | 0.26 |
| | | GLNR | GLN.x → GLN | 4.15E+01 | 4.06E+01 | 4.16E+01 | 1.43E+01 | 1.26E+01 | 1.94E+01 | 0.34 |
| | | GLUR | GLU ⇌ GLU.x | 1.62E+01 | 1.58E+01 | 1.68E+01 | 7.55E+00 | 6.88E+00 | 8.15E+00 | 0.47 |
| | | ASPR | ASP → ASP.x | 2.57E+00 | 2.53E+00 | 2.68E+00 | 1.08E+00 | 4.17E-01 | 1.69E+00 | 0.42 |
| | | SERR | SER.x → SER | 1.42E+01 | 1.35E+01 | 1.49E+01 | 5.49E+00 | 4.99E+00 | 6.06E+00 | 0.39 |
| | | CYSR | CYX.x → CYS +CYS | 4.41E+00 | 4.23E+00 | 4.58E+00 | 1.65E+00 | 1.32E+00 | 2.08E+00 | 0.37 |
| | | GLYR | GLY → GLY.x | 2.05E+00 | 1.90E+00 | 2.15E+00 | 2.60E-01 | 2.00E-02 | 4.92E-01 | 0.13 |
| | Glycolysis | HK | GLC → G6P | 5.14E+02 | 5.11E+02 | 5.21E+02 | 4.41E+02 | 4.26E+02 | 4.58E+02 | 0.86 |
| | | PGI | G6P ⇌ F6P | 5.11E+02 | 4.99E+02 | 5.24E+02 | 4.23E+02 | 4.04E+02 | 4.40E+02 | 0.83 |
| | | PFK | F6P ⇌ FBP | 5.09E+02 | 5.00E+02 | 5.12E+02 | 4.32E+02 | 4.17E+02 | 4.49E+02 | 0.85 |
| | | ALDO | FBP ⇌ DHAP +GAP | 5.09E+02 | 5.00E+02 | 5.12E+02 | 4.32E+02 | 4.17E+02 | 4.49E+02 | 0.85 |
| | | TPI | DHAP ⇌ GAP | 5.08E+02 | 5.06E+02 | 5.08E+02 | 4.31E+02 | 4.15E+02 | 4.48E+02 | 0.85 |
| | | GAPDH | GAP ⇌ 3 PG | 1.02E+03 | 9.96E+02 | 1.04E+03 | 8.69E+02 | 8.35E+02 | 9.03E+02 | 0.85 |
| | | ENO | 3 PG ⇌ PEP | 1.01E+03 | 9.99E+02 | 1.03E+03 | 8.68E+02 | 8.36E+02 | 9.00E+02 | 0.86 |
| | | PK | PEP → PYR.c | 1.04E+03 | 9.95E+02 | 1.04E+03 | 8.78E+02 | 8.36E+02 | 9.21E+02 | 0.84 |
| | | LDH | PYR.c ⇌ LAC | 9.99E+02 | 9.98E+02 | 1.02E+03 | 8.91E+02 | 8.62E+02 | 9.25E+02 | 0.89 |
| | | GPT1 | PYR.c ⇌ ALA | 1.19E+01 | 9.12E+00 | 1.19E+01 | 5.55E+00 | -9.08E+02 | 6.13E+00 | 0.47 |
| | | GPT2 | PYR.m ⇌ ALA | -2.58E+00 | -4.56E+00 | 2.87E+00 | -2.40E-03 | -3.22E+01 | 9.11E+02 | |
| | Pentose phosphate pathway | G6PD | G6P → P5P+CO2 | 1.26E-07 | 0.00E+00 | 3.91E-01 | 1.62E+01 | 4.41E+00 | 2.89E+01 | 128571428.57 |
| | | TK1 | P5P+P5 P ⇌ S7P+GAP | -9.11E-01 | -9.29E-01 | -8.30E-01 | 4.76E+00 | -1.22E-01 | 9.62E+00 | -5.23 |
| | | TA | S7P+GAP ⇌ F6P+E4 P | -9.11E-01 | -9.29E-01 | -8.30E-01 | 4.76E+00 | -1.22E-01 | 9.62E+00 | -5.23 |
| | | TK2 | P5P+E4 P ⇌ F6P+GAP | -9.11E-01 | -9.29E-01 | -8.30E-01 | 4.76E+00 | -1.22E-01 | 9.62E+00 | -5.23 |

*Table 1 continued on next page*

Table 1 continued

| Type | Pathway | ID | Reaction | 21% * | | | 0.5% † | | | Ratio |
|---|---|---|---|---|---|---|---|---|---|---|
| | | | | Flux | LB | UB | Flux | LB | UB | |
| | Anaplerosis | PYRT | PYR.c → PYR.m | 1.16E+02 | 1.16E+02 | 1.19E+02 | 4.42E+01 | 3.82E+01 | 9.58E+02 | |
| | | PC | PYR.m+CO2 → OAC | 1.88E+01 | 1.74E+01 | 1.91E+01 | 1.37E+01 | 9.82E+00 | 2.69E+01 | |
| | | PEPCK | OAC → PEP +CO2 | 2.56E+01 | 1.58E+01 | 2.57E+01 | 9.66E+00 | 0.00E+00 | 2.60E+01 | |
| | | ME2 | MAL → PYR.m +CO2 | 2.05E+00 | 9.51E-02 | 2.68E+00 | 1.00E-07 | 0.00E+00 | 2.25E+01 | |
| | | ME1 | MAL → PYR.c +CO2 | 2.78E-02 | 0.00E+00 | 2.63E+01 | 8.71E-05 | 0.00E+00 | 2.52E+01 | |
| | | FAO | FAO → AcCoA.m | 1.00E-07 | 0.00E+00 | 2.13E+00 | 6.58E-06 | 0.00E+00 | 7.73E-01 | |
| | | GLDH | GLU ↔ AKG | 1.71E+01 | 1.56E+01 | 1.84E+01 | 9.11E-01 | −6.16E-01 | 7.27E+00 | 0.05 |
| | | GLS | GLN → GLU | 3.78E+01 | 3.60E+01 | 3.86E+01 | 1.17E+01 | 1.01E+01 | 1.70E+01 | 0.31 |
| | Tricarboxylic acid cycle | PDH | PYR.m → AcCoA.m +CO2 | 1.02E+02 | 8.76E+01 | 1.15E+02 | 3.05E+01 | 2.86E+01 | 5.24E+01 | 0.3 |
| | | CS | AcCoA.m+OAC → CIT | 1.02E+02 | 8.30E+01 | 1.11E+02 | 3.05E+01 | 2.88E+01 | 5.09E+01 | 0.3 |
| | | IDH | CIT ↔ AKG +CO2 | 2.49E+01 | 2.42E+01 | 2.53E+01 | 1.01E+01 | 8.75E+00 | 1.41E+01 | 0.41 |
| | | OGDH | AKG → SUC +CO2 | 4.19E+01 | 4.01E+01 | 4.25E+01 | 1.10E+01 | 7.87E+00 | 2.02E+01 | 0.26 |
| | | SDH | SUC ↔ FUM | 4.19E+01 | 4.01E+01 | 4.25E+01 | 1.10E+01 | 7.87E+00 | 2.02E+01 | 0.26 |
| | | FH | FUM ↔ MAL | 4.19E+01 | 4.01E+01 | 4.25E+01 | 1.10E+01 | 7.87E+00 | 2.02E+01 | 0.26 |
| | | MDH | MAL ↔ OAC | 1.17E+02 | 1.08E+02 | 1.24E+02 | 3.14E+01 | 2.62E+01 | 5.70E+01 | 0.27 |
| | | GOT | OAC ↔ ASP | 8.11E+00 | 8.06E+00 | 8.23E+01 | 4.98E+00 | 4.32E+00 | 5.64E+00 | 0.61 |
| | | PST | 3 PG → SER | 1.95E+00 | 1.63E+00 | 2.00E+00 | 2.42E-01 | 1.34E-01 | 3.57E+01 | |
| | Amino acid metabolism | SHT | SER ↔ GLY +MEETHF | 6.38E+00 | 6.22E+00 | 6.43E+00 | 3.91E+00 | 3.71E+00 | 4.10E+00 | 0.61 |
| | | CYST | SER ↔ CYS | −7.12E+00 | −7.19E+00 | −6.81E+00 | −2.10E+00 | −2.97E+00 | −1.44E+00 | 0.3 |
| | | SD | SER → PYR.c | 1.17E+01 | 1.04E+01 | 1.20E+01 | 2.82E-01 | 0.00E+00 | 1.47E+00 | 0.02 |
| | | GLYS | CO2 +MEETHF → GLY | 3.39E+00 | 3.35E+00 | 3.49E+00 | 1.80E+00 | 1.66E+00 | 1.93E+00 | 0.53 |
| | Biomass | BIOMASS | 1216*AcCoA.c+295.6 *ALA +232.4*ASP +114.7*CO2 +71.43*CYS +57.14*DHAP +142.4*G6P+158.6 *GLN +190.1*GLU +324.2*GLY +125.6*MEETHF +114.7*PSP+217.2*SER → biomass | 2.38E-02 | 2.34E-02 | 2.39E-02 | 1.68E-02 | 1.61E-02 | 1.75E-02 | 0.71 |
| | | ACL | CIT → AcCoA.c +MAL | 7.74E+01 | 6.29E+01 | 1.04E+02 | 2.04E+01 | 1.95E+01 | 3.71E+01 | 0.26 |
| | | LIPS | AcCoA.c → lipid | 4.84E+01 | 4.55E+01 | 4.84E+01 | 1.00E-07 | 0.00E+00 | 1.68E+01 | 0 |

*Table 1 continued*

| Type | Pathway | ID | Reaction | 21% * | | | Flux | 0.5%† | | Ratio |
|---|---|---|---|---|---|---|---|---|---|---|
| | | | | Flux | LB | UB | | LB | UB | |
| | Mixing | cPYR | 0*PYR.c → PYR.ms | 1.00E+00 | 8.47E-01 | 1.00E+00 | 1.42E-01 | 0.00E+00 | 1.00E+00 | 1.00E+00 |
| | Mixing | mPYR | 0*PYR.m → PYR.ms | 1.00E-07 | 0.00E+00 | 1.53E-01 | 8.58E-01 | 0.00E+00 | 1.00E+00 | 1.00E+00 |
| | | sPYR | PYR.ms → PYR.fix | 1.00E+00 | 1.00E+00 | 1.00E+00 | 1.00E+00 | 1.00E+00 | 1.00E+00 | 1.00E+00 |
| | Transport | MCT | LAC ↔ LAC.x | 1.00E-07 | 0.00E+00 | 1.05E-01 | 1.52E+03 | 1.35E+03 | 2.41E+03 | 1520000000 |
| | Transport | GLUR | GLU ↔ GLU.x | 5.10E+00 | 4.77E+00 | 5.23E+00 | 1.54E+00 | 1.11E+00 | 2.54E+00 | 0.3 |
| | | PGI | G6P ↔ F6P | 2.78E+05 | 1.77E+05 | Inf | 2.46E+05 | 0.00E+00 | Inf | |
| | | ALDO | FBP ↔ DHAP +GAP | 1.43E+02 | 1.43E+02 | 1.43E+02 | 3.20E+02 | 2.79E+02 | 3.60E+02 | 2.24 |
| | | TPI | DHAP ↔ GAP | 4.33E+03 | 4.33E+03 | 1.09E+04 | 1.70E+03 | 1.06E+03 | 3.06E+03 | 0.39 |
| | Glycolysis | GAPDH | GAP ↔ 3 PG | 4.42E+02 | 4.72E+00 | 4.50E+02 | 1.00E-07 | 0.00E+00 | 2.39E+02 | |
| | | LDH | PYR.c ↔ LAC | 1.63E+03 | 1.62E+03 | 1.80E+03 | 4.80E+00 | 0.00E+00 | 3.51E+02 | 0 |
| | | GPT1 | PYR.c ↔ ALA | 1.00E-07 | 0.00E+00 | 2.61E-01 | 8.32E+02 | 0.00E+00 | 9.06E+02 | |
| | | GPT2 | PYR.m ↔ ALA | 4.21E-04 | 0.00E+00 | 2.92E+00 | 1.28E-04 | 0.00E+00 | | |
| | | TK1 | P5P+P5 P ↔ S7P+GAP | 9.97E+04 | 6.27E+03 | Inf | 1.47E+02 | 6.67E+01 | 2.60E+02 | 0 |
| EXCH | Pentose phosphate pathway | TA | S7P+GAP ↔ F6P+E4 P | 5.93E+00 | 5.79E+00 | 6.97E+00 | 2.35E-04 | 0.00E+00 | 7.54E+00 | |
| | | TK2 | P5P+E4 P ↔ F6P+GAP | 1.00E+07 | -Inf | Inf | 9.05E+00 | 4.10E+00 | 1.43E+01 | |
| | | GLDH | GLU ↔ AKG | 1.52E+03 | 1.52E+03 | 7.13E+03 | 3.78E+02 | 1.93E+02 | 1.94E+03 | |
| | Anaplerosis | GLS | GLN ↔ GLU | 3.99E-01 | 0.00E+00 | 8.04E-01 | 1.00E-07 | 0.00E+00 | 3.84E-01 | |
| | | IDH | CIT ↔ AKG +CO2 | 4.55E+00 | 4.03E+00 | 5.19E+00 | 2.52E+00 | 1.80E+00 | 4.50E+00 | |
| | | SDH | SUC ↔ FUM | 1.22E+03 | | Inf | 7.60E+01 | 2.57E+01 | Inf | |
| | Tricarboxylic acid cycle | FH | FUM ↔ MAL | 3.66E+05 | 1.95E+05 | Inf | 5.05E+05 | 3.06E+02 | Inf | |
| | | MDH | MAL ↔ OAC | 1.11E+03 | 7.88E+02 | 2.38E+03 | 1.33E+02 | 7.22E+01 | 3.25E+02 | 0.12 |
| | | GOT | OAC ↔ ASP | 1.00E+07 | -Inf | Inf | 4.42E+01 | 0.00E+00 | Inf | |
| | Amino acid metabolism | SHT | SER ↔ GLY +MEETHF | 5.10E+00 | 8.92E-01 | 5.25E+00 | 6.07E-07 | 0.00E+00 | 3.32E+02 | |
| | | CYST | SER ↔ CYS | 1.52E-05 | 0.00E+00 | 2.55E-04 | 1.46E-02 | 0.00E+00 | Inf | |

*SSR 391.7 [311.2–416.6] (95% CI, 362 DOF).
†SSR 334.3 [311.2–416.6] (95% CI, 362 DOF).

**Table 2.** LF fluxes following DMSO and BAY treatment.

| Type | Pathway | ID | Reaction | DMSO* | | | BAY† | | | Ratio |
|---|---|---|---|---|---|---|---|---|---|---|
| | | | | Flux | LB | UB | Flux | LB | UB | |
| Transport | | GLUT | GLC.x ↔ GLC | 6.12E+02 | 6.12E+02 | 6.12E+02 | 8.80E+02 | 8.80E+02 | 8.80E+02 | 1.44 |
| | | PYRR | PYR.x ↔ PYR.c | 9.98E+01 | 9.95E+01 | 1.01E+02 | 6.06E+01 | 6.06E+01 | 6.06E+01 | 0.61 |
| | | MCT | LAC ↔ LAC.x | 8.19E+02 | 8.17E+02 | 8.20E+02 | 1.33E+03 | 1.33E+03 | 1.33E+03 | 1.62 |
| | | ALAR | ALA → ALA.x | 2.67E+00 | 2.36E+00 | 3.29E+00 | 5.98E+00 | 5.88E+00 | 6.24E+00 | 2.24 |
| | | GLNR | GLN.x → GLN | 3.78E+01 | 3.77E+01 | 3.79E+01 | 2.06E+01 | 2.06E+01 | 2.06E+01 | 0.54 |
| | | GLUR | GLU → GLU.x | 1.61E+01 | 1.56E+01 | 1.62E+01 | 1.68E+01 | 1.68E+01 | 1.68E+01 | 1.05 |
| | | ASPR | ASP → ASP.x | 2.36E+00 | 2.32E+00 | 2.49E+00 | 1.80E+00 | 1.80E+00 | 1.81E+00 | 0.76 |
| | | SERR | SER.x → SER | 1.03E+01 | 1.03E+01 | 1.06E+01 | 2.50E+00 | 2.50E+00 | 2.50E+00 | 0.24 |
| | | CYSR | CYX.x ↔ CYS+CYS | 2.79E+00 | 2.79E+00 | 2.95E+00 | 3.07E-01 | 3.06E-01 | 3.07E-01 | 0.11 |
| | | GLYR | GLY → GLY.x | 2.52E+00 | 2.30E+00 | 2.73E+00 | 5.52E-01 | 4.30E-01 | 7.45E-01 | 0.22 |
| NET | Glycolysis | HK | GLC → G6P | 6.12E+02 | 6.12E+02 | 6.12E+02 | 8.80E+02 | 8.80E+02 | 8.80E+02 | 1.44 |
| | | PGI | G6P ↔ F6P | 6.09E+02 | 6.08E+02 | 6.09E+02 | 8.42E+02 | 8.42E+02 | 8.42E+02 | 1.38 |
| | | PFK | F6P → FBP | 6.07E+02 | 6.07E+02 | 6.07E+02 | 8.65E+02 | 8.65E+02 | 8.65E+02 | 1.43 |
| | | ALDO | FBP ↔ DHAP+GAP | 6.07E+02 | 6.07E+02 | 6.07E+02 | 8.65E+02 | 8.65E+02 | 8.65E+02 | 1.43 |
| | | TPI | DHAP ↔ GAP | 6.06E+02 | 6.06E+02 | 6.06E+02 | 8.65E+02 | 8.65E+02 | 8.65E+02 | 1.43 |
| | | GAPDH | GAP ↔ 3PG | 1.21E+03 | 1.21E+03 | 1.21E+03 | 1.74E+03 | 1.74E+03 | 1.74E+03 | 1.44 |
| | | ENO | 3PG ↔ PEP | 1.21E+03 | 1.21E+03 | 1.21E+03 | 1.57E+03 | 1.57E+03 | 1.57E+03 | 1.3 |
| | | PK | PEP → PYR.c | 1.23E+03 | 1.19E+03 | 1.23E+03 | 1.65E+03 | 1.65E+03 | 1.65E+03 | 1.34 |
| | | LDH | PYR.c ↔ LAC | 8.19E+02 | 8.17E+02 | 8.20E+02 | 1.33E+03 | 1.33E+03 | 1.33E+03 | 1.62 |
| | | GPT1 | PYR.c ↔ ALA | 9.62E+00 | 9.44E+00 | 9.62E+00 | 9.36E+00 | 9.32E+00 | 9.42E+00 | 0.97 |
| | | GPT2 | PYR.m ↔ ALA | 1.14E-01 | | | 2.28E-07 | -1.22E-05 | 6.41E-04 | |
| | Pentose phosphate pathway | G6PD | G6P → P5P+CO2 | 2.02E-02 | 0.00E+00 | 1.08E+00 | 3.64E+01 | 3.64E+01 | 3.64E+01 | 1801.98 |
| | | TK1 | P5P+P5P ↔ S7P+GAP | -9.06E-01 | -9.28E-01 | -9.06E-01 | 1.17E+01 | 1.17E+01 | 1.17E+01 | -12.89 |
| | | TA | S7P+GAP ↔ F6P+E4P | -9.06E-01 | -9.28E-01 | -9.06E-01 | 1.17E+01 | 1.17E+01 | 1.17E+01 | -12.89 |
| | | TK2 | P5P+E4P ↔ F6P+GAP | -9.06E-01 | -9.28E-01 | -9.06E-01 | 1.17E+01 | 1.17E+01 | 1.17E+01 | -12.89 |
| | Anaplerosis | PYRT | PYR.c → PYR.m | 4.99E+02 | 4.97E+02 | 4.99E+02 | 5.50E+02 | 5.50E+02 | 5.50E+02 | 1.1 |
| | | PC | PYR.m+CO2 → OAC | 2.11E+01 | 2.07E+01 | 2.17E+01 | 9.05E+01 | 9.05E+01 | 9.05E+01 | 4.28 |
| | | PEPCK | OAC → PEP+CO2 | 1.36E+01 | 1.36E+01 | 1.37E+01 | 8.58E+01 | 8.58E+01 | 8.58E+01 | 6.31 |
| | | ME2 | MAL → PYR.m+CO2 | 1.30E+01 | 1.28E+01 | 1.37E+01 | 1.00E-07 | 0.00E+00 | 9.49E-06 | 0 |
| | | ME1 | MAL → PYR.c+CO2 | 3.20E-03 | 0.00E+00 | 1.73E+00 | 1.00E-07 | 0.00E+00 | 2.15E-05 | |
| | | FAO | FAO → AcCoA.m | 1.00E-07 | 0.00E+00 | 3.48E+00 | 1.09E-04 | 8.34E-06 | 4.14E-02 | |
| | | GLDH | GLU ↔ AKG | 1.33E+01 | 1.31E+01 | 1.35E+01 | -2.46E-01 | -2.47E-01 | -2.46E-01 | -0.02 |
| | | GLS | GLN ↔ GLU | 3.40E+01 | 3.35E+01 | 3.42E+01 | 1.88E+01 | 1.88E+01 | 1.88E+01 | 0.55 |

*Table 2 continued on next page*

Table 2 continued

| Type | Pathway | ID | Reaction | DMSO* | | | BAY† | | | |
|---|---|---|---|---|---|---|---|---|---|---|
| | | | | Flux | LB | UB | Flux | LB | UB | Ratio |
| | | PDH | PYR.m → AcCoA.m+CO2 | 4.90E+02 | 4.90E+02 | 4.92E+02 | 4.60E+02 | 4.60E+02 | 4.60E+02 | 0.94 |
| | | CS | AcCoA.m+OAC → CIT | 4.90E+02 | 4.84E+02 | 4.91E+02 | 4.60E+02 | 4.60E+02 | 4.60E+02 | 0.94 |
| | | IDH | CIT → AKG+CO2 | 2.70E+01 | 2.70E+01 | 2.76E+01 | 1.45E+01 | 1.45E+01 | 1.45E+01 | 0.54 |
| | Tricarboxylic acid cycle | OGDH | AKG → SUC+CO2 | 4.03E+01 | 3.99E+01 | 4.04E+01 | 1.43E+01 | 1.43E+01 | 1.43E+01 | 0.35 |
| | | SDH | SUC → FUM | 4.03E+01 | 3.99E+01 | 4.04E+01 | 1.43E+01 | 1.43E+01 | 1.43E+01 | 0.35 |
| | | FH | FUM → MAL | 4.03E+01 | 3.99E+01 | 4.04E+01 | 1.43E+01 | 1.43E+01 | 1.43E+01 | 0.35 |
| | | MDH | MAL → OAC | 4.91E+02 | 4.91E+02 | 4.92E+02 | 4.60E+02 | 4.60E+02 | 4.60E+02 | 0.94 |
| | | GOT | OAC → ASP | 7.91E+00 | 7.76E+00 | 7.98E+00 | 4.46E+00 | 4.46E+00 | 4.46E+00 | 0.56 |
| | | PST | 3PG → SER | 4.03E-01 | 3.74E-01 | 5.04E-01 | 1.73E-02 | 1.73E-02 | 1.73E-02 | 429.83 |
| | | SHT | SER → GLY+MEETHF | 6.63E+00 | 6.59E+00 | 6.65E+00 | 2.85E+00 | 2.79E+00 | 2.93E+00 | 0.43 |
| | Amino acid metabolism | CYST | SER → CYS | -3.88E+00 | -3.91E+00 | -3.87E+00 | 2.03E-01 | 2.02E-01 | 2.03E-01 | -0.05 |
| | | SD | SER → PYR.c | 2.80E+00 | 2.80E+00 | 2.80E+00 | 1.70E+02 | 1.70E+02 | 1.70E+02 | 60.81 |
| | | GLYS | CO2+MEETHF → GLY | 3.63E+00 | 3.50E+00 | 3.65E+00 | 1.41E+00 | 1.30E+00 | 1.46E+00 | 0.39 |
| | Biomass | BIOMASS | 1216*AcCoA.c+295.6 *ALA +232.4*ASP+114.7*CO2+71.43*CYS+57.14*DHAP+142.4*G6P+ 158.6*GLN+190.1*GLU +324.2*GLY+125.6*MEETHF+114.7*P5P+217.2 *SER → biomass | 2.39E-02 | 2.39E-02 | 2.50E-02 | 1.14E-02 | 1.14E-02 | 1.14E-02 | 0.48 |
| | | ACL | CIT → AcCoA.c+MAL | 4.63E+02 | 4.63E+02 | 4.66E+02 | 4.45E+02 | 4.45E+02 | 4.45E+02 | 0.96 |
| | | LIPS | AcCoA.c → lipid | 4.34E+02 | 4.29E+02 | 4.34E+02 | 4.32E+02 | 4.32E+02 | 4.32E+02 | |
| | | cPYR | 0*PYR.c → PYR.ms | 1.00E+00 | 9.99E-01 | 1.00E+00 | 1.00E-07 | 0.00E+00 | 1.00E+00 | |
| | Mixing | mPYR | 0*PYR.m → PYR.ms | 1.00E-07 | 0.00E+00 | 9.83E-04 | 1.00E+00 | 0.00E+00 | 1.00E+00 | |
| | | sPYR | PYR.ms → PYR.fix | 1.00E+00 | 1.00E+00 | 1.00E+00 | 1.00E+00 | 1.00E+00 | 1.00E+00 | |

Table 2 continued on next page

*Table 2 continued*

| Type | Pathway | ID | Reaction | DMSO* | | | BAY† | | | Ratio |
|---|---|---|---|---|---|---|---|---|---|---|
| | | | | Flux | LB | UB | Flux | LB | UB | |
| EXCH | Transport | MCT | LAC ↔ LAC.x | 6.24E-04 | 0.00E+00 | 3.56E+00 | 7.11E+02 | 7.11E+02 | 7.11E+02 | 1139423.08 |
| | | GLUR | GLU ↔ GLU.x | 5.06E+00 | 4.82E+00 | 5.75E+00 | 3.48E+00 | 3.48E+00 | 3.48E+00 | 0.69 |
| | | PGI | G6P ↔ F6P | 1.40E+06 | 1.39E+06 | Inf | 4.31E+06 | 4.31E+06 | 4.31E+06 | |
| | | ALDO | FBP ↔ DHAP+GAP | 2.38E+02 | 2.38E+02 | 2.38E+02 | 1.02E+03 | 1.02E+03 | 1.02E+03 | 4.28 |
| | | TPI | DHAP ↔ GAP | 9.99E+06 | | Inf | 7.57E+03 | 7.57E+03 | 7.57E+03 | |
| | Glycolysis | GAPDH | GAP ↔ 3PG | 5.81E+02 | 5.81E+02 | 7.25E+02 | 1.09E+02 | 1.07E+02 | 1.09E+02 | 0.19 |
| | | LDH | PYR.c ↔ LAC | 2.65E+03 | 2.58E+03 | 2.65E+03 | 4.92E+01 | 4.91E+01 | 4.94E+01 | 0.02 |
| | | GPT1 | PYR.c ↔ ALA | 1.00E-07 | 0.00E+00 | 5.60E-02 | 2.45E+03 | 2.45E+03 | 2.45E+03 | 24500000000 |
| | | GPT2 | PYR.m ↔ ALA | 1.00E-07 | 0.00E+00 | 5.65E-02 | 1.00E-07 | 0.00E+00 | 1.20E-05 | |
| | Pentose phosphate pathway | TK1 | P5P+P5P ↔ S7P+GAP | 1.28E+06 | 9.01E+03 | Inf | 1.00E+07 | -Inf | Inf | |
| | | TA | S7P+GAP ↔ F6P+E4P | 8.89E+00 | 8.88E+00 | 9.53E+00 | 5.10E+01 | 5.10E+01 | 5.10E+01 | 5.74 |
| | | TK2 | P5P+E4P ↔ F6P+GAP | 6.93E+00 | 5.12E+00 | 6.98E+00 | 1.00E-07 | 0.00E+00 | 1.56E-04 | 0 |
| | Anaplerosis | GLDH | GLU ↔ AKG | 5.63E+03 | 4.43E+03 | 5.66E+03 | 1.42E+03 | 1.42E+03 | 1.42E+03 | 0.25 |
| | | GLS | GLN ↔ GLU | 1.27E+00 | 1.20E+00 | 1.50E+00 | 5.52E-01 | 5.51E-01 | 5.55E-01 | 0.43 |
| | | IDH | CIT ↔ AKG+CO2 | 3.36E+00 | 3.24E+00 | 3.92E+00 | 4.66E+00 | 4.66E+00 | 4.66E+00 | 1.39 |
| | | SDH | SUC ↔ FUM | 4.30E+02 | 4.30E+02 | 1.46E+06 | 1.04E+04 | 1.04E+04 | 1.04E+04 | |
| | Tricarboxylic acid cycle | FH | FUM ↔ MAL | 7.29E+06 | -Inf | Inf | 4.56E+06 | 4.56E+06 | 4.56E+06 | |
| | | MDH | MAL ↔ OAC | 5.49E+02 | 5.47E+02 | 5.49E+02 | 1.00E-07 | 0.00E+00 | 6.30E-03 | 0 |
| | | GOT | OAC ↔ ASP | 1.04E+02 | 1.04E+02 | 1.04E+02 | 4.76E+05 | 4.76E+05 | 4.76E+05 | 4576.92 |
| | Amino acid metabolism | SHT | SER ↔ GLY+MEETHF | 1.39E+00 | 1.37E+00 | 1.41E+00 | 1.86E+03 | 1.86E+03 | 1.86E+03 | 1338.13 |
| | | CYST | SER ↔ CYS | 1.25E-07 | 0.00E+00 | 4.22E-02 | 1.33E-01 | 1.33E-01 | 1.33E-01 | 1064000 |

*SSR 393.5 [311.2–416.6] (95% CI, 362 DOF).

†SSR 392.4 [308.4–413.4] (95% CI, 359 DOF).

**Table 3.** PASMC fluxes in 21% and 0.5% oxygen.

| Type | Pathway | ID | Reaction | 21%* | | | 0.5%† | | | Ratio |
|---|---|---|---|---|---|---|---|---|---|---|
| | | | | Flux | LB | UB | Flux | LB | UB | |
| NET | Transport | GLUT | GLC.x → GLC | 4.28E+02 | 4.28E+02 | 4.28E+02 | 3.65E+02 | 3.65E+02 | 3.65E+02 | 0.85 |
| | | PYRR | PYR.x → PYR.c | 1.04E+02 | 1.02E+02 | 1.09E+02 | 4.53E+01 | 4.31E+01 | 4.57E+01 | 0.44 |
| | | MCT | LAC ↔ LAC.x | 8.01E+02 | 8.01E+02 | 8.04E+02 | 6.49E+02 | 6.49E+02 | 6.49E+02 | 0.81 |
| | | ALAR | ALA → ALA.x | 1.43E+01 | 1.43E+01 | 1.46E+01 | 7.83E+00 | 7.83E+00 | 8.24E+00 | 0.55 |
| | | GLNR | GLN.x → GLN | 7.73E+01 | 7.53E+01 | 7.73E+01 | 1.77E+02 | 1.77E+02 | 1.77E+02 | 2.29 |
| | | GLUR | GLU ↔ GLU.x | 2.53E+01 | 2.52E+01 | 2.54E+01 | 1.19E+01 | 1.19E+01 | 1.22E+01 | 0.47 |
| | | ASPR | ASP → ASP.x | 7.01E+00 | 6.99E+00 | 7.02E+00 | 6.92E+00 | 6.84E+00 | 7.00E+00 | |
| | | SERR | SER.x → SER | 2.54E+00 | 2.48E+00 | 2.55E+00 | 2.57E+00 | 2.55E+00 | 2.57E+00 | 1.01 |
| | | CYSR | CYX.x → CYS +CYS | 6.39E+00 | 6.34E+00 | 6.45E+00 | 3.75E+00 | 3.75E+00 | 3.75E+00 | 0.59 |
| | | GLYR | GLY → GLY.x | 3.66E-01 | 3.03E-01 | 4.19E-01 | 4.06E-01 | 3.86E-01 | 4.25E-01 | |
| | Glycolysis | HK | GLC → G6P | 4.28E+02 | 4.28E+02 | 4.28E+02 | 3.65E+02 | 3.65E+02 | 3.65E+02 | 0.85 |
| | | PGI | G6P ↔ F6P | 4.06E+02 | 4.06E+02 | 4.07E+02 | 3.62E+02 | 3.62E+02 | 3.63E+02 | 0.89 |
| | | PFK | F6P → FBP | 4.17E+02 | 4.17E+02 | 4.18E+02 | 3.61E+02 | 3.60E+02 | 3.61E+02 | 0.87 |
| | | ALDO | FBP ↔ DHAP +GAP | 4.17E+02 | 4.17E+02 | 4.18E+02 | 3.61E+02 | 3.60E+02 | 3.61E+02 | 0.87 |
| | | TPI | DHAP ↔ GAP | 4.16E+02 | 4.16E+02 | 4.16E+02 | 3.60E+02 | 3.60E+02 | 3.60E+02 | 0.87 |
| | | GAPDH | GAP ↔ 3 PG | 8.39E+02 | 8.39E+02 | 8.41E+02 | 7.21E+02 | 7.21E+02 | 7.21E+02 | 0.86 |
| | | ENO | 3 PG → PEP | 8.36E+02 | 8.35E+02 | 8.53E+02 | 7.20E+02 | 7.20E+02 | 7.20E+02 | 0.86 |
| | | PK | PEP → PYR.c | 9.31E+02 | 9.30E+02 | 9.31E+02 | 9.24E+02 | 9.24E+02 | 9.24E+02 | 0.99 |
| | | LDH | PYR.c ↔ LAC | 8.01E+02 | 8.01E+02 | 8.04E+02 | 6.49E+02 | 6.49E+02 | 6.49E+02 | 0.81 |
| | | GPT1 | PYR.c ↔ ALA | 1.64E+02 | 1.62E+02 | 1.92E+02 | −1.36E+01 | −1.39E+01 | −1.35E+01 | −0.08 |
| | | GPT2 | PYR.m ↔ ALA | −1.43E+02 | −1.43E+02 | −1.42E+02 | 2.62E+01 | 2.51E+01 | 2.65E+01 | −0.18 |
| | Pentose phosphate pathway | G6PD | G6P → P5P+CO2 | 1.89E+01 | 1.57E+01 | 1.93E+01 | 1.16E-07 | 0.00E+00 | 1.10E-03 | 0 |
| | | TK1 | P5P+P5 P ↔ S7P+GAP | 5.46E+00 | 4.44E+00 | 5.96E+00 | −6.15E-01 | −6.15E-01 | −5.77E-01 | −0.11 |
| | | TA | S7P+GAP ↔ F6P+E4 P | 5.46E+00 | 4.44E+00 | 5.96E+00 | −6.15E-01 | −6.15E-01 | −5.77E-01 | −0.11 |
| | | TK2 | P5P+E4 P ↔ F6P+GAP | 5.46E+00 | 4.44E+00 | 5.96E+00 | −6.15E-01 | −6.15E-01 | −5.77E-01 | −0.11 |
| | Anaplerosis | PYRT | PYR.c → PYR.m | 7.60E+01 | 7.59E+01 | 7.66E+01 | 3.36E+02 | 3.36E+02 | 3.36E+02 | 4.42 |
| | | PC | PYR.m+CO2 → OAC | 6.30E+01 | 6.29E+01 | 6.59E+01 | 2.37E+02 | 2.36E+02 | 2.37E+02 | 3.76 |
| | | PEPCK | OAC → PEP +CO2 | 9.51E+01 | 9.51E+01 | 9.53E+01 | 2.03E+02 | 2.03E+02 | 2.04E+02 | 2.14 |
| | | ME2 | MAL → PYR.m +CO2 | 1.20E-03 | 0.00E+00 | 5.20E-03 | 1.82E+02 | 1.81E+02 | 1.82E+02 | 151517.08 |
| | | ME1 | MAL → PYR.c +CO2 | 3.29E-05 | 0.00E+00 | 1.15E+00 | 5.91E-05 | 0.00E+00 | 8.06E-02 | |
| | | FAO | FAO → AcCoA.m | 1.00E-07 | 0.00E+00 | 1.32E-02 | 1.15E-04 | 0.00E+00 | 1.56E-01 | |
| | | GLDH | GLU ↔ AKG | 4.43E+01 | 4.42E+01 | 4.45E+01 | 1.59E+02 | 1.59E+02 | 1.59E+02 | 3.6 |
| | | GLS | GLN ↔ GLU | 7.38E+01 | 7.36E+01 | 7.38E+01 | 1.74E+02 | 1.74E+02 | 1.74E+02 | 2.36 |

*Table 3 continued on next page*

Table 3 continued

| Type | Pathway | ID | Reaction | 21%* | | | 0.5%† | | | Ratio |
|---|---|---|---|---|---|---|---|---|---|---|
| | | | | Flux | LB | UB | Flux | LB | UB | |
| | Tricarboxylic acid cycle | PDH | PYR.m → AcCoA.m +CO2 | 1.56E+02 | 1.48E+02 | 1.66E+02 | 2.55E+02 | 2.55E+02 | 2.55E+02 | 1.63 |
| | | CS | AcCoA.m+OAC → CIT | 1.56E+02 | 1.56E+02 | 1.58E+02 | 2.55E+02 | 2.55E+02 | 2.55E+02 | 1.63 |
| | | IDH | CIT ↔ AKG +CO2 | 2.11E+01 | 2.10E+01 | 2.11E+01 | 2.16E+01 | 2.16E+01 | 2.16E+01 | 1.03 |
| | | OGDH | AKG → SUC +CO2 | 6.54E+01 | 6.51E+01 | 6.59E+01 | 1.81E+02 | 1.80E+02 | 1.81E+02 | 2.77 |
| | | SDH | SUC ↔ FUM | 6.54E+01 | 6.51E+01 | 6.59E+01 | 1.81E+02 | 1.80E+02 | 1.81E+02 | 2.77 |
| | | FH | FUM ↔ MAL | 6.54E+01 | 6.51E+01 | 6.59E+01 | 1.81E+02 | 1.80E+02 | 1.81E+02 | 2.77 |
| | | MDH | MAL ↔ OAC | 2.01E+02 | 2.01E+02 | 2.01E+02 | 2.32E+02 | 2.32E+02 | 2.33E+02 | 1.16 |
| | | GOT | OAC ↔ ASP | 1.22E+01 | 1.17E+01 | 1.24E+01 | 1.07E+01 | 1.06E+01 | 1.07E+01 | 0.87 |
| | Amino acid metabolism | PST | 3 PG → SER | 2.69E+00 | 2.57E+00 | 2.80E+00 | 7.12E-01 | 7.01E-01 | 7.21E-01 | 0.26 |
| | | SHT | SER ↔ GLY +MEETHF | 5.19E+00 | 5.15E+00 | 5.20E+00 | 3.82E+00 | 3.81E+00 | 3.86E+00 | 0.74 |
| | | CYST | SER ↔ CYS | −1.12E+01 | −1.17E+01 | −1.11E+01 | −6.35E+00 | −6.35E+00 | −6.35E+00 | 0.57 |
| | | SD | SER → PYR.c | 6.39E+00 | 6.23E+00 | 6.44E+00 | 2.33E+00 | 2.33E+00 | 2.33E+00 | 0.36 |
| | | GLYS | CO2 +MEETHF → GLY | 2.39E+00 | 2.36E+00 | 2.42E+00 | 1.80E+00 | 1.79E+00 | 1.81E+00 | 0.75 |
| | Biomass | BIOMASS | 978*AcCoA.c+237.8*ALA +187*ASP +92.3*CO2 +57.46*CYS +45.97*DHAP +114.5*G6P+127.6*GLN +153*GLU +260.8*GLY +101.1*MEETHF +92.3*P5P+174.8*SER → biomass | 2.77E-02 | 2.70E-02 | 2.79E-02 | 2.00E-02 | 2.00E-02 | 2.00E-02 | 0.72 |
| | | ACL | CIT → AcCoA.c +MAL | 1.35E+02 | 1.34E+02 | 1.38E+02 | 2.33E+02 | 2.33E+02 | 2.33E+02 | 1.72 |
| | | LIPS | AcCoA.c → lipid | 1.08E+02 | 9.99E+01 | 1.08E+02 | 2.14E+02 | 2.14E+02 | 2.14E+02 | 1.98 |
| | Mixing | cPYR | 0*PYR.c → PYR.ms | 5.77E-01 | 5.64E-01 | 5.92E-01 | 1.00E+00 | 9.96E-01 | 1.00E+00 | 1.73 |
| | | mPYR | 0*PYR.m → PYR.ms | 4.23E-01 | 4.08E-01 | 4.36E-01 | 1.00E-07 | 0.00E+00 | 4.40E-03 | 0 |
| | | sPYR | PYR.ms → PYR.fix | 1.00E+00 | 1.00E+00 | 1.00E+00 | 1.00E+00 | 1.00E+00 | 1.00E+00 | |

*Table 3 continued*

| Type | Pathway | ID | Reaction | 21%* | | | 0.5%† | | | Ratio |
|---|---|---|---|---|---|---|---|---|---|---|
| | | | | Flux | LB | UB | Flux | LB | UB | |
| EXCH | Transport | MCT | LAC ↔ LAC.x | 1.00E-07 | 0.00E+00 | 1.36E+02 | 1.64E+03 | 1.63E+03 | 1.65E+03 | 16400000000 |
| | | GLUR | GLU ↔ GLU.x | 1.00E-07 | 0.00E+00 | 2.27E-02 | 5.69E-05 | 0.00E+00 | 1.71E-02 | |
| | Glycolysis | PGI | G6P ↔ F6P | 4.88E+06 | 4.88E+06 | Inf | 9.92E+06 | 9.85E+04 | Inf | |
| | | ALDO | FBP ↔ DHAP +GAP | 2.89E+02 | 2.80E+02 | 2.89E+02 | 2.57E+02 | 2.56E+02 | 2.57E+02 | 0.89 |
| | | TPI | DHAP ↔ GAP | 9.86E+06 | -Inf | Inf | 1.65E+03 | 1.63E+03 | 1.68E+03 | |
| | | GAPDH | GAP ↔ 3 PG | 1.12E+03 | 0.00E+00 | 5.88E+05 | 1.00E-07 | 0.00E+00 | 2.27E-01 | |
| | | LDH | PYR.c ↔ LAC | 1.47E+03 | 1.39E+03 | 1.47E+03 | 4.49E+02 | 4.49E+02 | 4.49E+02 | 0.31 |
| | | GPT1 | PYR.c ↔ ALA | 2.74E+02 | 2.73E+02 | 2.77E+02 | 1.00E-07 | 0.00E+00 | 4.28E-02 | 0 |
| | | GPT2 | PYR.m ↔ ALA | 1.38E+02 | 1.38E+02 | 1.49E+02 | 9.64E+01 | 0.00E+00 | 1.01E+02 | 0.7 |
| | Pentose phosphate pathway | TK1 | P5P+P5 P ↔ S7P+GAP | 7.99E+02 | 7.97E+02 | 8.08E+02 | 3.54E+01 | 3.54E+01 | 3.55E+01 | 0.04 |
| | | TA | S7P+GAP ↔ F6P+E4 P | 1.53E-01 | 0.00E+00 | 5.82E-01 | 2.55E+00 | 2.54E+00 | 2.57E+00 | 16.67 |
| | | TK2 | P5P+E4 P ↔ F6P+GAP | 3.33E+00 | 2.62E+00 | 3.35E+00 | 1.29E+01 | 1.29E+01 | 1.29E+01 | 3.88 |
| | Anaplerosis | GLDH | GLU ↔ AKG | 5.36E+02 | 5.34E+02 | 8.37E+02 | 1.23E+03 | 1.23E+03 | 1.23E+03 | 2.29 |
| | | GLS | GLN ↔ GLU | 3.20E-01 | 0.00E+00 | 2.74E+00 | 1.12E+00 | 1.07E+00 | 1.74E+00 | |
| | Tricarboxylic acid cycle | IDH | CIT ↔ AKG +CO2 | 1.04E+01 | 1.02E+01 | 1.04E+01 | 6.30E+01 | 6.30E+01 | 6.31E+01 | 6.09 |
| | | SDH | SUC ↔ FUM | 2.78E-01 | 0.00E+00 | Inf | 3.34E+06 | 3.34E+06 | 3.34E+06 | |
| | | FH | FUM ↔ MAL | 1.03E-04 | 0.00E+00 | 1.58E+01 | 2.18E+02 | 2.18E+02 | 2.18E+02 | 2114238.83 |
| | | MDH | MAL ↔ OAC | 1.01E+03 | 8.27E+02 | 1.01E+03 | 3.67E+03 | 3.67E+03 | 3.69E+03 | 3.63 |
| | | GOT | OAC ↔ ASP | 2.27E+02 | 2.27E+02 | 2.47E+02 | 1.54E+01 | 1.54E+01 | 1.55E+01 | 0.07 |
| | Amino acid metabolism | SHT | SER ↔ GLY +MEETHF | 3.55E+00 | 3.52E+00 | 3.59E+00 | 1.60E-01 | 1.36E-01 | 1.70E-01 | 0.05 |
| | | CYST | SER ↔ CYS | 1.04E+03 | 1.03E+03 | 1.04E+03 | 2.00E-03 | 0.00E+00 | 2.0E-03 | 0 |

*SSR 575.6 [499.1–630.6] (95% CI, 563 DOF).

†SSR 521.3 [482.2–611.6] (95% CI, 545 DOF).

consistent with prior reports describing hypoxia-mediated increases in gluconeogenesis and glycogen synthesis (*Favaro et al., 2012*; *Owczarek et al., 2020*; *Pelletier et al., 2012*). These data suggest that lactate also makes a small (~5% carbon) contribution to glycogen precursors. Together, these findings from stable isotope tracing of lactate reveal its important contribution to primary cell metabolism under standard culture conditions, and also reveal increased lactate uptake in hypoxia.

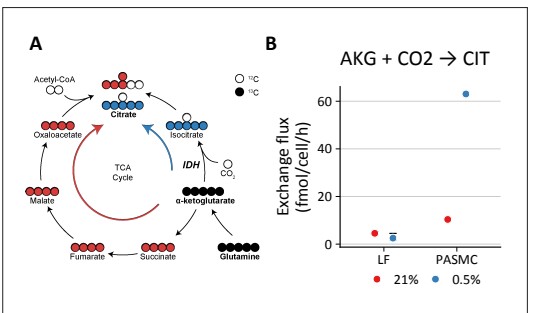

**Figure 6.** Hypoxia increases reductive carboxylation in pulmonary artery smooth muscle cells. (**A**) Reductive carboxylation describes the converstion of α-ketoglutarate (AKG) to isocitrate by reverse flux through isocitrate dehydrogenase (IDH) (*blue* arrow). This yields M+5 citrate. (**B**) Exchange flux estimates for reductive carboxylation. Data show the model estimate with upper and lower bounds for LFs and PASMCs.

## Hypoxia prevents BAY from increasing glycolysis

To reconcile the differential effects of PHD inhibition by hypoxia and BAY, we next addressed whether hypoxia could suppress BAY-stimulated glucose and lactate fluxes (*Figure 8*). LFs cultured in standard growth medium were treated with BAY and placed in either 21% or 0.5% oxygen. Similar to previous experiments, BAY decreased cell growth rate, increased glucose uptake, and increased lactate efflux in 21% oxygen. However, when combined with 0.5% oxygen, BAY did not enhance lactate efflux. These data demonstrate that hypoxia antagonizes the effects of HIF-1α activation on glycolytic flux.

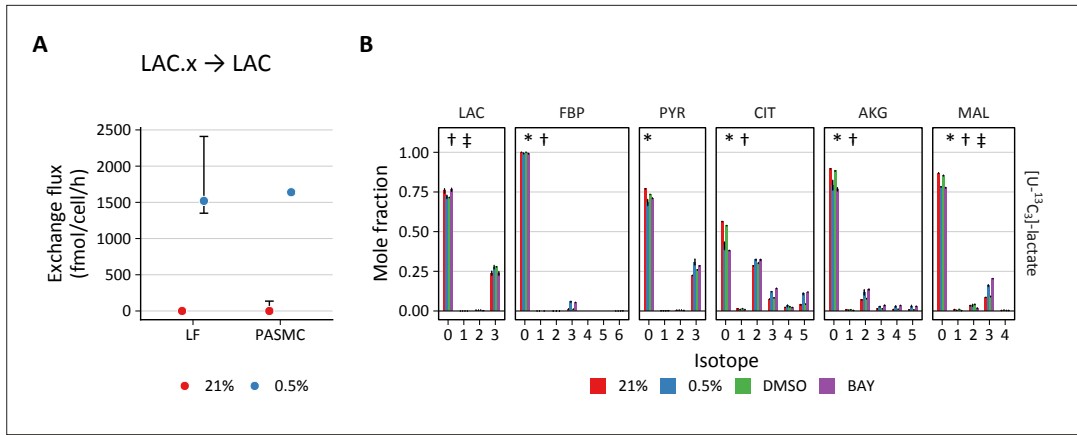

**Figure 7.** PHD inhibition increases lactate uptake and oxidation. (**A**) Exchange flux estimates for lactate import presented as in (**A**). (**C**) Mass isotopomer distributions of lactate (LAC), fructose-bisphosphate (FBP), pyruvate (PYR), citrate (CIT), α-ketoglutarate (AKG), and malate (MAL) in LFs following 72 hr labeling with [U-$^{13}$C$_3$] lactate. Data are mean ± SEM (n=4, p<0.05 for the following comparisons: * 0.5% *v.* 21% oxygen, † BAY *v.* DMSO, ‡ Δoxygen *v.* ΔBAY).

We observed a similar effect of hypoxia on glycolysis when using RNA interference to silence PHD2 expression and activate HIF-1α gene transcription (*Figure 8—figure supplement 1*). As with BAY treatment, siPHD2 stabilized HIF-1α in normoxia and increased glycolytic gene expression (*Figure 8— figure supplement 1B–E*); however, hypoxia inhibited siPHD2-dependent increases in glycolysis (*Figure 8—figure supplement 1F*). These data indicate that the metabolic consequences of BAY treatment are a direct consequence of PHD inhibition and not simply an off-target effect.

To investigate these metabolic differences further, we performed metabolomic profiling of LFs treated for 72 hr with hypoxia or BAY separately or in combination. Both 0.5% oxygen and BAY treatment caused marked changes in intracellular metabolites (*Figure 9—figure supplement 1*). Of 133 total metabolites, 98 were differentially regulated by hypoxia and 54 were differentially regulated by BAY. Of these, 44 were modulated by both treatments (*Figure 9—figure supplement 1C*). Metabolite set enrichment analysis of KEGG biochemical pathways identified increased enrichment of fatty acid biosynthesis pathways with hypoxia (*Figure 9—figure supplement 1D*). By contrast, BAY treatment was enriched for metabolites involved in pentose/glucuronate interconversions and glycolysis

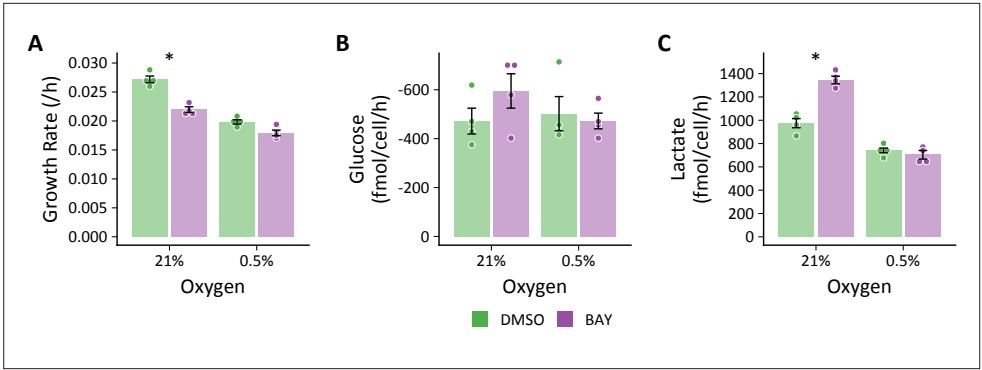

**Figure 8.** Hypoxia prevents BAY-stimulated increases in glycolysis. (**A**) LFs were cultured in standard growth medium and treated with molidustat (BAY, 10 μM) or vehicle (DMSO, 0.1%) in 21% or 0.5% oxygen conditions for 72 hr. Growth rates were determined by linear fitting of log-transformed growth curves. (**B**) Glucose uptake in LFs treated with BAY cultured in hypoxia (note reversed *y*-axis). (**C**) Lactate efflux in LFs treated with BAY cultured in hypoxia. Data are mean ± SEM (n=4, * p<0.05 BAY *v.* DMSO within a given oxygen exposure).

The online version of this article includes the following source data and figure supplement(s) for figure 8:

**Figure supplement 1.** Hypoxia attenuates siPHD2-stimulated increases in glycolysis.

**Figure supplement 1—source data 1.** Uncropped blot images for *Figure 8—figure supplement 1*.

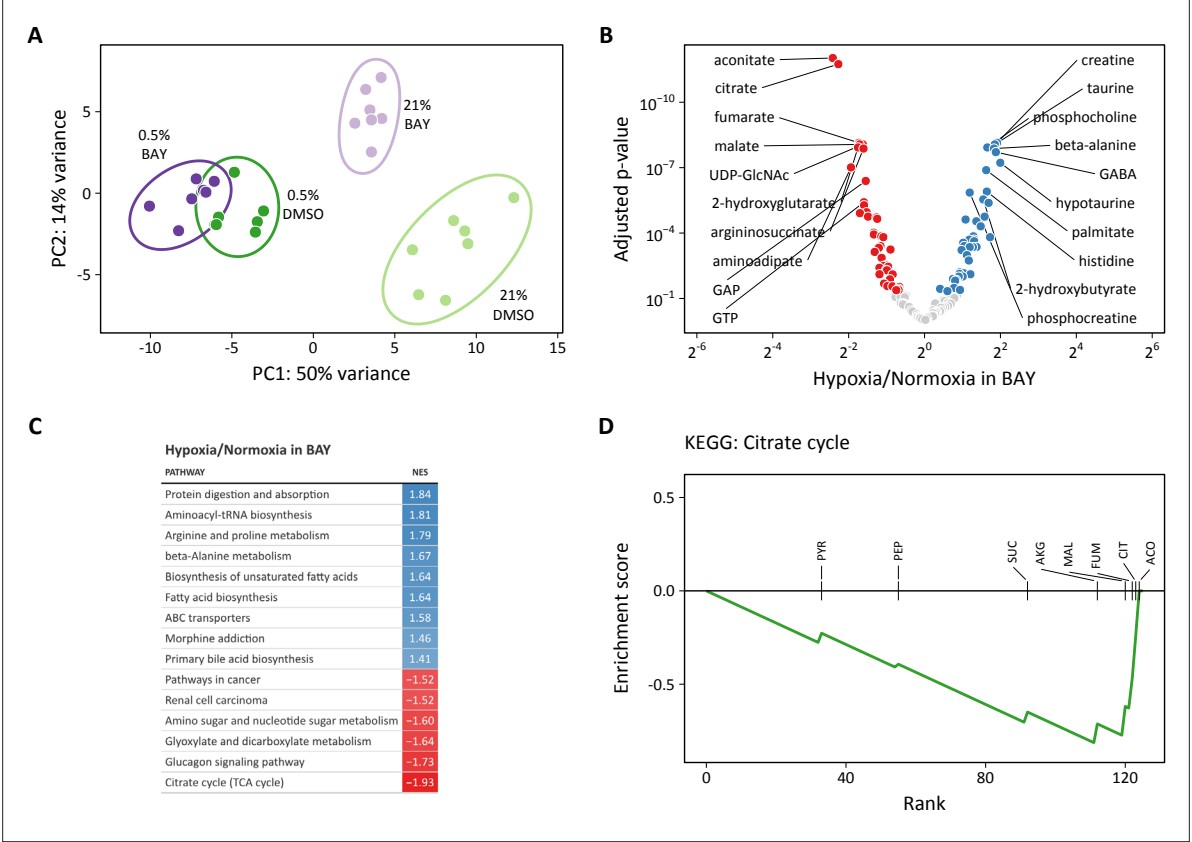

**Figure 9.** Metabolomic analysis of molidustat treatment in normoxia and hypoxia. (**A**) Principal component analysis of intracellular metabolites following 72 hr of treatment suggests a dominant effect of hypoxia over pharmacologic PHD inhibition on the metabolome (n=7). (**B**) Volcano plot of intracellular metabolites of BAY-treated cells cultured in 21% or 0.5% oxygen. Filled circles indicate significantly increased (*blue*) and decreased (*red*) levels in hypoxia. (**C**) Metabolite set enrichment analysis of metabolites from (**B**). All KEGG pathways with p-values <0.05 are shown. (**D**) Leading edge analysis of the TCA cycle metabolite set. Negative values indicate the relative enrichment associated with BAY treatment alone compared to BAY plus hypoxia treatment. Abbreviations: PYR, pyruvate; SUC, succinate; PEP, phosphoenolpyruvate; CIT, citrate; AKG, α-ketoglutarate; MAL, malate; ACO, aconitate; FUM, fumarate.

The online version of this article includes the following source data and figure supplement(s) for figure 9:

**Source data 1.** Source data for *Figure 9C*.

**Figure supplement 1.** Metabolomic profiling of hypoxia and BAY treated lung fibroblasts.

**Figure supplement 1—source data 1.** Source data for *Figure 9—figure supplement 1D*.

**Figure supplement 1—source data 2.** Source data for *Figure 9—figure supplement 1E*.

**Figure supplement 2.** Changes in NAD(H) and NADP(H) following hypoxia and BAY treamtent.

(*Figure 9—figure supplement 1E*). Aspartate was the most significantly decreased metabolite with both treatments, consistent with prior reports demonstrating an important role for HIF-1 regulation of aspartate biosynthesis in cancer cells (*Garcia-Bermudez et al., 2018*; *Meléndez-Rodríguez et al., 2019*).

Principal component analysis revealed greater similarity among both treatment groups cultured in 0.5% oxygen than among the BAY-treatment groups (*Figure 9A*). Moreover, these hypoxia-treated cells were well-segregated from cells treated with BAY alone. These observations are consistent with our metabolic flux models demonstrating an overriding effect of hypoxia per se on cell metabolism and highlighting important differences between hypoxic and pharmacologic PHD inhibition. To identify the metabolic changes that depend on hypoxia rather than PHD inhibition, we identified differentially regulated metabolites in BAY-treated cells cultured in normoxia and hypoxia (*Figure 9B*). Of 133 metabolites, 83 were significantly differentially regulated by hypoxia in BAY-treated cells. An enrichment analysis of these metabolites demonstrated up-regulation of arginine and proline

metabolism and down-regulation of the TCA cycle as the most impacted by hypoxia in BAY treated cells (*Figure 9C*). Leading edge analysis highlights negative enrichment scores associated with all of the TCA metabolites detected by our platform (*Figure 9D*). This result indicates better preservation of TCA cycle flux in normoxic BAY-treated cells than in hypoxic cells, as suggested by our metabolic flux models where hypoxia caused a 1.5- to 2-fold reduction of TCA flux compared to a 1.1–1.5-fold reduction with BAY treatment (*Figure 5*).

In addition to these differential effects on polar metabolite levels, we reasoned that another critical difference between hypoxia and BAY treatment is the impact of hypoxia on cellular redox state. We measured the impact of these treatments on intracellular NAD(H) and NADP(H) redox couples (*Figure 9—figure supplement 2*). Hypoxia increased the NADH/NAD$^+$ ratio, driven primarily by a decrease in intracellular NAD$^+$. Interestingly, while BAY treatment increased the levels of NADH, a concomitant increase in NAD$^+$ resulted in preservation of the NADH/NAD$^+$ ratio. As NADH accumulation is a putative inhibitor of glycolytic flux (*Tilton et al., 1991*), this may be one mechanism by which glycolytic flux is decreased in hypoxia but not following BAY treatment. Conversely, hypoxia decreased the NADPH/NADP$^+$ ratio in both control and BAY-treated cells, primarily through decreasing NADPH. This finding is consistent with prior studies of bovine coronary artery smooth muscle cells (*Gupte and Wolin, 2006*). In hypoxia, NADPH is generated primarily by the pentose phosphate pathway (*Liu et al., 2016*), where it plays an important role in the detoxification of reactive oxygen species (*Fessel and Oldham, 2018*; *Xiao et al., 2018*). Although we modeled increased flux through the pentose phosphate pathway in hypoxia (*Figure 5A*), the overall flux was low and apparently inadequate to maintain the NADPH/NADP$^+$ ratio.

## Transcriptomic analysis identifies metabolic regulators in hypoxia

To identify the upstream regulators of the observed metabolic changes, we next performed RNA-seq transcriptomic analysis of LFs treated with hypoxia or BAY, separately or together. As anticipated, both hypoxia and BAY treatment induced substantial changes in gene expression (*Figure 10—figure supplement 1*). Of the 10,686 differentially expressed genes across both conditions, 869 (4%) were unique to BAY treatment in normoxia, 4002 (19%) were shared by BAY and hypoxia, while 5052 (25%) were unique to 0.5% hypoxia (*Figure 10—figure supplement 1C*). Gene set enrichment analysis of these differentially regulated metabolites was performed using Molecular Signatures Database 'Hallmark' gene sets (*Liberzon et al., 2015*; *Figure 10—figure supplement 1D–F*). As expected, both treatments were associated with enrichment of the 'hypoxia' and 'glycolysis' gene sets.

Given the disparate effects of hypoxic and pharmacologic PHD inhibition on cellular metabolism described above, we focused our transcriptomic analyses on the differences between hypoxia and BAY treatments. Principal component analysis again demonstrated clear separation among the four treatment groups (*Figure 10A*). The first and second principal components correspond to 0.5% oxygen and BAY treatments, respectively. Consistent with our prior observations, the combination of 0.5% oxygen plus BAY was more similar to 0.5% oxygen alone than BAY treatment alone, supporting the hypothesis that hypoxia overrides the effects of BAY treatment. To identify the transcripts driving these differences, we identified genes differentially expressed following hypoxia in BAY-treated cells (*Figure 10B*). An enrichment analysis of these differentially expressed genes identified pro-proliferative gene sets like 'E2F targets', 'G2/M checkpoint', and 'MYC targets' associated with hypoxia (*Figure 10C–E*). These findings were further supported by a transcription factor enrichment analysis identifying enrichment of MYC transcription factor targets associated with hypoxia, but not BAY treatment (*Figure 10F*, *Figure 10—figure supplement 1G–H*).

As one example of MYC-dependent transcriptional regulation, MYC represses the transcription of *CDKN1A*, which encodes the cyclin-dependent kinase and cell-cycle inhibitor p21 (*García-Gutiérrez et al., 2019*). *CDKN1A* expression increases with BAY treatment in normoxia, but is decreased by hypoxia, consistent with hypoxia-induced MYC activation (*Figure 10—figure supplement 2A*). While we did not observe increased *CDKN1A* expression by RT-qPCR in siPHD2-treated cells in normoxia, we did observe hypoxia-mediated down-regulation in both siCTL- and siPHD2-treated cells (*Figure 10—figure supplement 2B*), consistent with MYC transcriptional activity.

Classically, hypoxia and HIF activation are thought to inhibit cell proliferation by inhibiting pro-proliferative MYC signaling (*Koshiji et al., 2004*). These results indicate that hypoxia-induced MYC

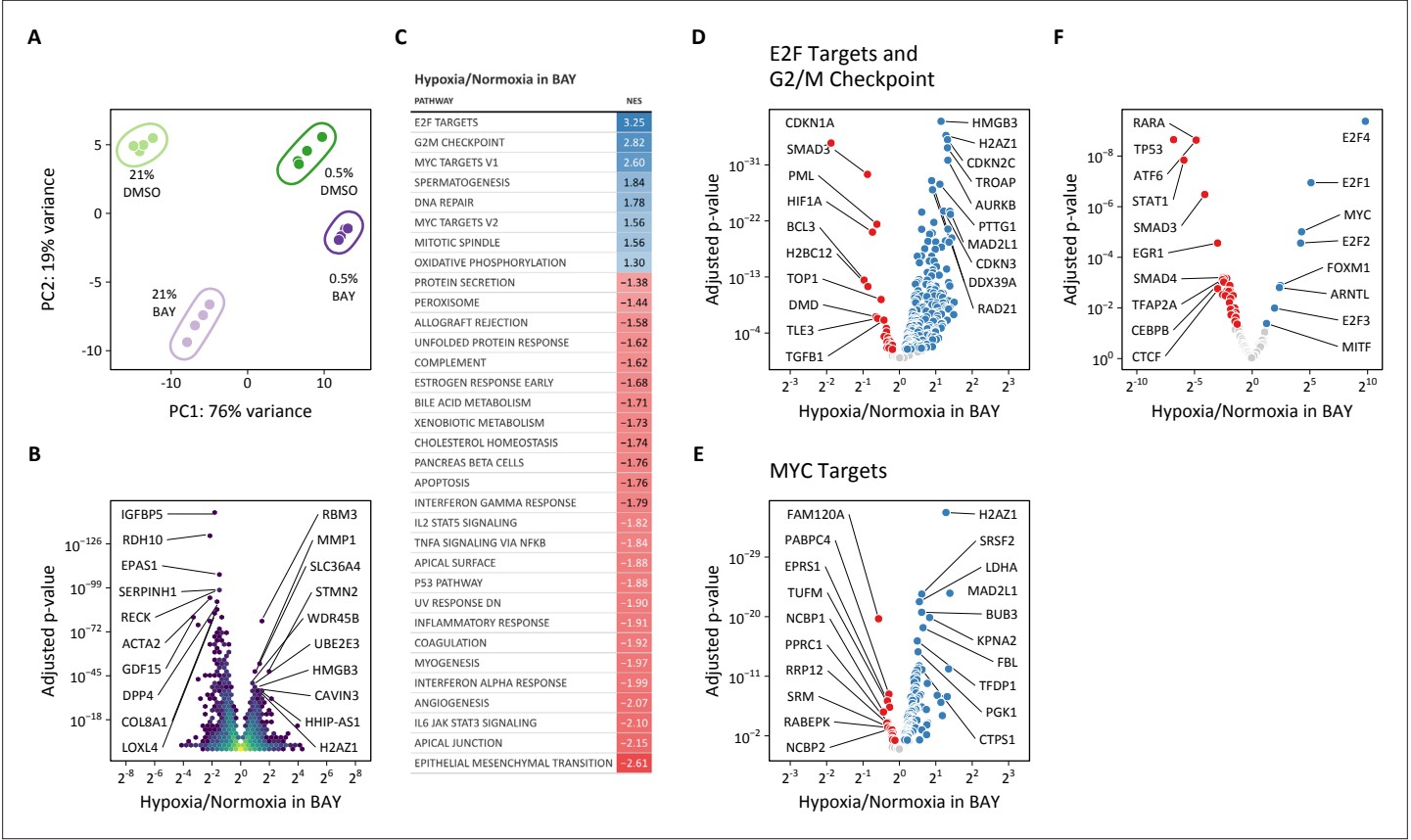

**Figure 10.** Transcriptomic analysis of molidustat treatment in normoxia and hypoxia. (**A**) Principal component analysis of transcriptional changes in lung fibroblasts following 72 hr of treatment with 0.5% oxygen or molidustat (BAY), separately or together (n=4). (**B**) Volcano plot illustrating the effects of hypoxia in BAY-treated cells on gene expression. (**C**) Gene set enrichment analysis of transcripts from (**B**). (**D**) Volcano plot of those transcripts comprising the E2F Targets and G2/M Checkpoint Hallmark gene sets. (**E**) Volcano plot of those transcripts comprising the MYC Targets V1 and V2 Hallmark gene sets. (**F**) Volcano plot illustrating the results of a transcription factor enrichment analysis suggests mechanisms for differential regulation of gene expression following hypoxia or BAY treatment.

The online version of this article includes the following source data and figure supplement(s) for figure 10:

**Source data 1.** Source data for *Figure 10C*.

**Figure supplement 1.** Transcriptomic profiling of hypoxia and BAY-treated lung fibroblasts.

**Figure supplement 1—source data 1.** Source data for *Figure 10—figure supplement 1E*.

**Figure supplement 1—source data 2.** Source data for *Figure 10—figure supplement 1F*.

**Figure supplement 2.** *CDKN1A* expression in BAY- and siPHD2-treated cells.

activation may be sustaining proliferation in LFs. We reasoned that these pro-proliferative signals may also account for the unexpected effects of hypoxia on glycolysis that we observed.

## MYC antagonizes HIF-dependent glycolytic fluxes

To examine the role of hypoxia-induced MYC activation in the metabolic response to hypoxia in proliferating primary cells, we first measured MYC by immunoblot. Consistent with our bioinformatic results, we observed increased MYC in hypoxia-treated cells, but not with BAY-treatment alone, where MYC was decreased (*Figure 11A–B*). These findings were similar in siPHD2-treated cells (*Figure 11— figure supplement 1*). Together, these data suggest a model whereby hypoxia activates MYC and inhibits HIF-driven increases in glycolysis (*Figure 11C*, **top**). Inhibiting MYC in hypoxia, therefore, should increase glycolysis. Conversely, HIF-dependent glycolysis after BAY treatment in normoxia should be sensitive to inhibition by MYC over-expression (*Figure 11C*, **bottom**).

To test the hypothesis that hypoxia-induced MYC expression inhibits glycolysis in primary cells, we first combined MYC knockdown with hypoxia treatment (*Figure 11C*, **top**). As expected, MYC-deficient

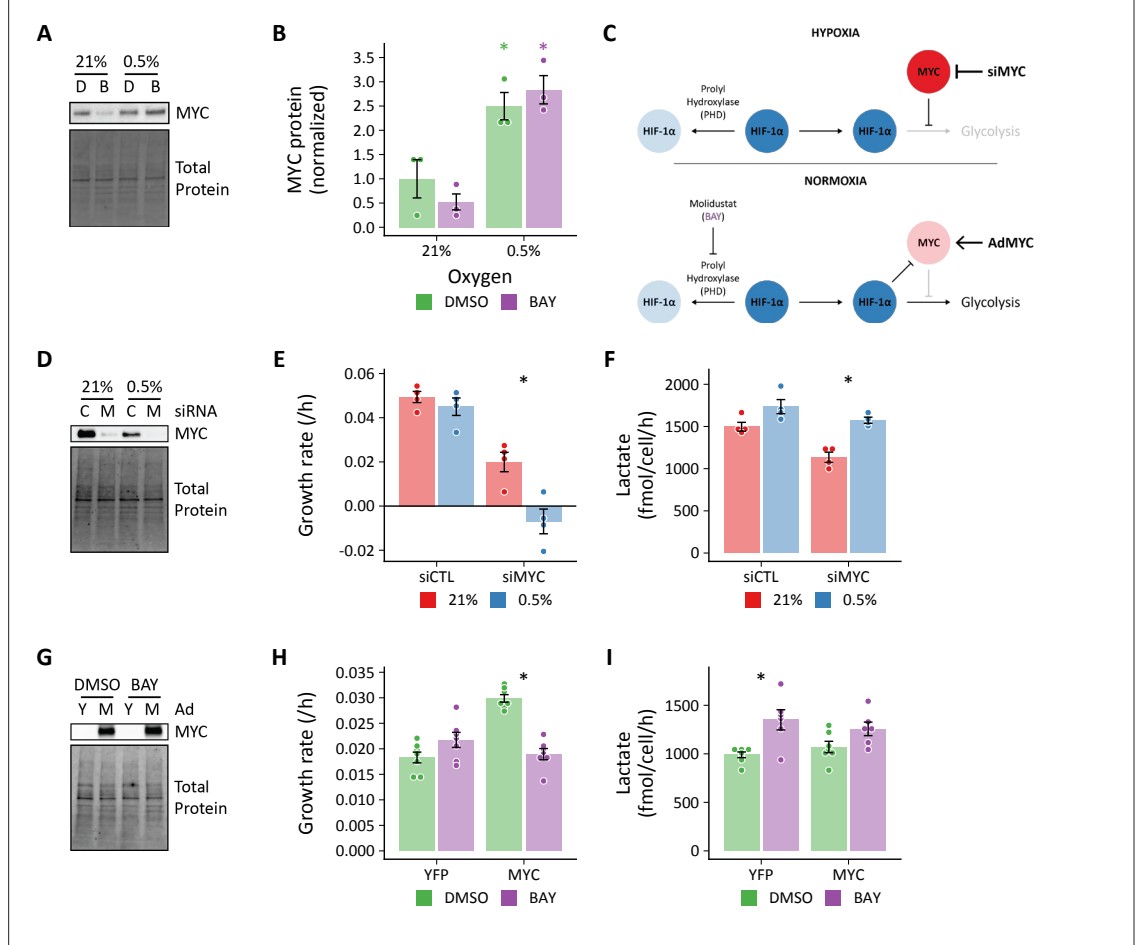

**Figure 11.** MYC regulates HIF-dependent glycolytic flux. (**A**) Representative immunoblot of MYC protein expression in lung fibroblasts (LFs) following 72 hr of treatment with 0.5% oxygen or molidustat BAY, (**B**). (**B**) Quantification of band densities from (**A**). (**C**) Working hypotheses: (1) Hypoxia-induced MYC antagonizes HIF-dependent metabolic and transcriptional events (*upper panel*). Silencing MYC with siRNA (siMYC) should therefore increase lactate efflux in hypoxia. (2) HIF activation in normoxia decreases MYC activity and increases glycolysis (*lower panel*). Therefore, overexpressing MYC (AdMYC) should restore MYC signaling and inhibit lactate efflux. (**D**) Representative immunoblot of LFs treated with siRNA targeting MYC (**M**) demonstrating adequate protein knockdown. (**E**) Growth rates of MYC-knockdown cells cultured in hypoxia. (**F**) Lactate efflux rates of MYC-knockdown cells cultured in hypoxia. (**G**) Representative immunoblot of LFs treated with MYC adenovirus. (**H**) Growth rates of MYC overexpressing cells cultured with BAY. (**I**) Lactate efflux rates of MYC overexpressing cells cultured with BAY. Data are mean ± SEM (n=3–7, p<0.05 as indicated by *black* * for differences within groups and *colored* * for differences between groups defined by the x-axis).

The online version of this article includes the following source data and figure supplement(s) for figure 11:

**Source data 1.** Uncropped blot images for *Figure 11A*.

**Source data 2.** Uncropped blot images for *Figure 11D and G*.

**Figure supplement 1.** Effects of hypoxia on MYC protein in siPHD2-treated cells.

**Figure supplement 1—source data 1.** Uncropped blot images for *Figure 11—figure supplement 1*.

**Figure supplement 2.** Effects of MYC on HIF stabilization and target gene exprssion.

**Figure supplement 2—source data 1.** Uncropped blot images for *Figure 11—figure supplement 2*.

cells proliferated more slowly in normoxia, and MYC was essential for sustaining cell proliferation in hypoxia (***Figure 11E***). We did not observe any significant differences in HIF-1α stabilization or target gene expression with MYC knockdown in normoxia or hypoxia (***Figure 11—figure supplement 2A–E***). Consistent with our hypothesis, MYC-knockdown increased lactate efflux in hypoxia, unlike control siRNA-treated cells (***Figure 11F***).

We next performed the complementary experiment to determine whether MYC over-expression could attenuate the increase in glycolysis observed with BAY treatment in normoxia (***Figure 11C***,

**bottom**). We did not observe any significant differences in cell growth rate with MYC over-expression (*Figure 11H*). Interestingly, MYC over-expression potentiated BAY-stimulated increases in HIF-1α stabilization and target gene expression (*Figure 11—figure supplement 2F–J*). In spite of these transcriptional changes, MYC over-expression attenuated BAY-stimulated lactate efflux in normoxia. Together, these data indicate that hypoxia-induced MYC expression may be one factor that uncouples the HIF transcriptional program from glycolytic flux in proliferating primary cells.

## Discussion

In this work, we used $^{13}$C metabolic flux analysis to identify hypoxia-mediated metabolic changes in proliferating human primary cells. Our principal finding was that hypoxia reduced, rather than increased, carbon flux through glycolysis and lactate fermentation pathways despite robust activation of the HIF-1α transcriptional program and up-regulation of glycolytic genes. The LFs studied here are certainly capable of augmenting glycolysis in response to HIF-1α stabilization, as demonstrated by experiments with the PHD inhibitor BAY; however, these effects are completely attenuated when BAY-treated cells are cultured in hypoxia. Together, these findings suggest that changes in enzyme expression alone are insufficient to alter metabolic flux in hypoxia and point to the importance of regulatory mechanisms that supersede the effects of HIF-dependent gene transcription in primary cells.

Our data indicate that hypoxia-induced MYC expression is one such regulatory mechanism. MYC is a transcription factor that regulates the expression of numerous genes involved in many biological processes, including metabolism, proliferation, apoptosis, and differentiation (*Dang et al., 2006*; *Li et al., 2020*; *Stine et al., 2015*). As deregulated MYC activity has been associated with the majority of human cancers (*Vita and Henriksson, 2006*), much of our understanding of MYC regulation comes from studies using cancer cell models (*Dang, 2012*; *Li et al., 2020*; *Madden et al., 2021*; *Stine et al., 2015*). The role of MYC in the biology of untransformed cells is less well understood. The literature describes a complex and reciprocal relationship between HIF and MYC that depends on both environmental (e.g. hypoxia) and cellular context (*Li et al., 2020*). Generally, HIF-1 has been observed to inhibit MYC through multiple mechanisms (*Gordan et al., 2007*; *Koshiji et al., 2005*; *Koshiji et al., 2004*; *Zhang et al., 2007*), and this previous work is consistent with our present observations that HIF stabilization following BAY treatment in normoxia decreased MYC protein and target gene expression. Conversely, MYC has been implicated in increased HIF activity through transcriptional and post-transcriptional mechanisms (*Chen et al., 2013*; *Doe et al., 2012*; *Zhang et al., 2009*), primarily in the context of malignant transformation. The observation that MYC may antagonize the transcriptional effects of HIF to sustain primary cell proliferation and metabolism in hypoxia suggests a substantially different regulatory relationship than has been previously described.

Understanding how MYC transcriptional activity affects hypoxic primary cell metabolism is imperative to our understanding of cellular adaptation to hypoxia. MYC stimulates the expression of nuclear-encoded mitochondrial genes and promotes mitochondrial biogenesis, both directly and through activation of mitochondrial transcriptional factor A (TFAM) (*Li et al., 2005*). Indeed, we found that the oxidative phosphorylation gene set was enriched in BAY-treated cells cultured in hypoxia (*Figure 10C*). In this way, hypoxic MYC activation may sustain energy production by oxidative phosphorylation, thereby decreasing the energetic demands that would otherwise drive increased glycolytic flux. Beyond oxidative phosphorylation, MYC regulates genes involved in many other intermediary metabolic pathways, including amino acids, nucleotides, and lipids (*Stine et al., 2015*), that may also impact the central pathways of carbon metabolism studied in this work.

Given the importance of MYC in regulating many aspects of cell biology, its intracellular concentration is closely regulated at multiple levels (*Stine et al., 2015*), including chromatin decompaction (*Devaiah et al., 2016*); gene transcription (*Spencer and Groudine, 1990*; *Vita and Henriksson, 2006*); mRNA stability (*Bernstein et al., 1992*; *Lemm and Ross, 2002*; *Weidensdorfer et al., 2009*) and translation (*Wall et al., 2008*); and protein degradation (*Adhikary and Eilers, 2005*; *Farrell and Sears, 2014*). MYC mRNA has a short half-life of less than 20 min (*Dani et al., 1984*) and hypoxia decreases MYC mRNA decay in immortalized cells by varied mechanisms (*Carraway et al., 2017*; *Fortenbery et al., 2018*; *Fry et al., 2017*; *Zhang et al., 2019*; *Zhang et al., 2017*). In HEK293T cells, 1% oxygen culture increased the half-life of MYC mRNA (*Fortenbery et al., 2018*). The authors also show that stabilizing HIF with pharmacologic prolyl hydroxylase inhibitors did not reproduce this effect. This differential effect of hypoxic and pharmacologic PHD inhibition resembles our data

here describing metabolic differences between these conditions. Electron transport chain inhibitors and ebselen, an antioxidant and mimic of glutathione peroxidase, prevented the hypoxia-mediated increase in MYC mRNA half-life. These experiments suggest an important role for mitochondrially derived reactive oxygen species (ROS) in regulating MYC mRNA stability. Hypoxia also increases N6-methyladenosine modification of MYC mRNA, which decreases ribonucleoprotein binding. Less ribonucleoprotein binding may also contribute to MYC mRNA stabilization (*Carraway et al., 2017*; *Fortenbery et al., 2018*). MYC mRNA stability may also be enhanced by the hypoxia-induced down-regulation of micro RNAs miR-449a-5p and let-7g, which contribute to hypoxia-induced proliferation of PASMCs in vitro and pulmonary hypertension in vivo (*Zhang et al., 2019*; *Zhang et al., 2017*). MYC protein also has a short half-life of less than 30 min (*Salghetti et al., 1999*) and is degraded by the proteasome (*Lutterbach and Hann, 1994*). Two phosphorylation sites, S62 and T58, regulate its stability MYC stability. S62 phosphorylation promotes MYC stability (*Lutterbach and Hann, 1994*; *Sears et al., 2000*) and T58 phosphorylation by GSK-3β or BRD4 (*Devaiah et al., 2020*; *Gregory et al., 2003*) leading to S62 dephosphorylation, ubiquitination, and proteolysis. Identifying which of these many regulatory mechanisms are most important for sustaining MYC signaling in hypoxia will be critically important for understanding primary cell biology in variety of physiologic and pathophysiologic contexts.

Beyond MYC, the identification of other HIF-independent mechanisms regulating primary cell adaption to hypoxia is of critical importance. Cells express several oxygen-dependent enzymes in addition to PHD whose activities may be altered in hypoxia but not by PHD inhibition. For example, PHD is one of many α-ketoglutarate-dependent dioxygenase enzymes that rely on molecular oxygen for their catalytic activity (*Islam et al., 2018*). Jumonji-C (JmjC) domain-containing histone demethylases are other prominent members of this family whose inhibition by hypoxia has been shown to cause rapid and HIF-independent induction of histone methylation (*Batie et al., 2019*). Similarly, a recently described cysteamine dioxygenase has been shown to mediate the oxygen-dependent degradation of Regulators of G protein Signaling 4 and 5 and IL-32 (*Masson et al., 2019*). In addition to dioxygenase enzymes, electron transport chain dysfunction resulting from impaired Complex IV activity leads to increased ROS production in hypoxia (*Chandel et al., 1998*). Mitochondrial ROS increase the half-lives of several mRNAs in hypoxia, including MYC, independent of HIF stabilization (*Guzy et al., 2005*). Finally, hypoxia imposes a reductive stress on cells associated with an increase in the NADH/NAD$^+$ ratio secondary to impaired electron transport (*Figure 9—figure supplement 2A–C*; *Chance and Williams, 1955*; *Garofalo et al., 1988*). NADH accumulation may slow glycolysis via feedback inhibition of GAPDH (*Tilton et al., 1991*). Any or all these molecular mechanisms may also contribute to uncoupling glycolytic enzyme expression from glycolytic flux as observed in the experiments described here.

In addition to its effects on cellular metabolism, another canonical role of HIF-1 activation is slowing of cellular proliferation rate in the face of limited oxygen availability (*Hubbi and Semenza, 2015*). The effects of HIF-1 on cell proliferation rate are mediated, in part, by increased expression of cyclin-dependent kinase inhibitor p21 (*CDKN1A*), inhibition of E2F targets (*Gardner et al., 2001*), and inhibition of pro-proliferative MYC signaling (*Koshiji et al., 2004*). These transcriptional effects are precisely what we observed in BAY treated LFs in normoxia. By contrast, hypoxia culture was associated with decreased expression of p21, consistent with a previous report (*Mizuno et al., 2009*), as well as increased expression of MYC protein and enrichment of MYC target genes. Indeed, the most marked differences between hypoxia and BAY treatment on LF gene transcription were the up-regulation of pro-proliferative gene sets containing E2F targets and G2/M checkpoint proteins. Much of this transcriptional response may be driven by hypoxia-induced up-regulation of MYC, which is known to stimulate cell cycle progression through its effects on the expression and activity of cyclins, cyclin-dependent kinases, and cyclin-dependent kinase inhibitors (*Hydbring et al., 2017*). Clarifying the complex interactions among HIFs, MYC, and cell proliferation will be important for understanding the cellular response of mesenchymal cells to tissue injury.

Taken together, these findings raise important questions regarding the cell-autonomous role of HIFs in the hypoxia response. On an organismal level, HIFs drive expression of angiogenic and erythropoietic factors to increase oxygen delivery to hypoxic tissues. Within individual cells, HIF-1α seems to be important for mitigating the adverse effects of ROS formation by dysfunctional electron transport in the mitochondria. Indeed, hypoxia increased oxygen consumption

and ROS production in HIF-1α-null mouse embryonic fibroblasts (MEFs), which was associated with increased cell death (*Zhang et al., 2008*). Interestingly, these cells also had increased ATP levels compared to wild type, suggesting that mitochondrial function was adequate under 1% oxygen culture conditions to support oxidative phosphorylation and to meet the energy needs of the cells. Given the prominence of HIFs in mediating the transcriptional response to hypoxia, it is somewhat surprising that neither PHD, HIFs, nor their downstream targets were found to be selectively essential as a function of oxygen tension in a genome-wide CRISPR growth screen of K562 human lymphoblasts cultured in normoxia or hypoxia (*Jain et al., 2020*). Similarly, knockout of HIF signaling did not affect growth, internal metabolite concentrations, glucose consumption, or lactate production under hypoxia by human acute myeloid leukemia cells (*Wierenga et al., 2019*). Together with our results, these studies highlight the need for additional research linking hypoxia-induced metabolic changes to their transcriptional and post-transcriptional regulatory mechanisms, particularly in primary cells.

This work also highlights two specific metabolic features that appear to be important in the metabolic response of primary cells to hypoxia. First, both LFs and PASMCs demonstrated notable incorporation of lactate-derived carbon into intracellular metabolic pathways that increased with hypoxia and BAY treatments. This finding is consistent with increasing evidence suggesting an important role for lactate as a metabolic fuel in several organ systems (*Faubert et al., 2013*; *Hui et al., 2017*). Although typically considered a metabolic waste product (*Rabinowitz and Enerbäck, 2020*), an important contribution of lactate *import* in supporting metabolic homeostasis in the face of an ischemic insult, which is associated with increased extracellular lactate, is an evolutionarily attractive hypothesis that merits further investigation. Second, PASMCs, but not LFs, demonstrated significant rates of reductive carboxylation that increased in 0.5% oxygen. Reductive carboxylation was first identified in hypoxic tumor cells where stable isotope tracing revealed $^{13}$C incorporation from labeled glutamine into lipids (*Gameiro et al., 2013*; *Metallo et al., 2011*; *Scott et al., 2011*; *Wise et al., 2011*). Hypoxia drives PASMC proliferation in vivo, contributing to the development pulmonary hypertension in humans and animal models. Isocitrate dehydrogenase has previously been implicated in the pathobiology of this disease (*Fessel et al., 2012*), and our findings suggest that reductive carboxylation catalyzed by isocitrate dehydrogenase may be a metabolic vulnerability of hypoxic PASMCs associated with pulmonary vascular disease.

Our finding that hypoxia was associated with decreased glycolysis and lactate fermentation was unexpected. Several aspects of our experimental design may have contributed to this finding. First, our goal was to understand how metabolic reprogramming may support cell proliferation in hypoxia. Thus, we measured metabolite fluxes in cells during the exponential growth phase accounting for cell growth rate, metabolite degradation rates, and medium evaporation with multiple measurements over a 72 hr time course. Often, cells are studied near confluence, where metabolic contributions to biomass production are less and the rate of glycolysis in hypoxia may be higher. Second, we began our experimental treatments 24 hr prior to collecting samples to ensure that the hypoxia metabolic program was established prior to labeling. Similar studies (*Grassian et al., 2014*; *Metallo et al., 2011*) typically placed cells into hypoxia at the time of labeling. Third, and perhaps most importantly, these flux determinations were performed in human primary cell cultures rather than immortalized cell lines. Although both cell types used in this study were derived from lung, we anticipate that many of our findings will be generalizable to primary cells from different tissues.

In summary, in this metabolic flux analysis of proliferating human primary cells in vitro, we have demonstrated that MYC uncouples an increase in HIF-dependent glycolytic gene transcription from glycolytic flux in hypoxia. Indeed, the degree of metabolic reprogramming in hypoxia was modest and suggests close coupling between proliferation and metabolism. In light of our findings, additional studies are warranted to clarify the role of HIFs in mediating the metabolic response to hypoxia, to determine how MYC activity is regulated by hypoxia, and to identify other key regulators of hypoxic metabolic reprogramming in primary cells. Moreover, these data strongly caution investigators against drawing conclusions about metabolite flux from measures of gene transcription alone.

## Materials and methods

**Key resources table**

| Reagent type (species) or resource | Designation | Source or reference | Identifiers | Additional information |
|---|---|---|---|---|
| Antibody | anti-HIF-1α (Mouse monoclonal) | BD Biosciences | 610958 | 1:1000 |
| Antibody | anti-c-MYC (Rabbit monoclonal) | Cell Signaling Technologies | D84C12 | 1:1000 |
| Antibody | anti-LDHA (Rabbit polyclonal) | Cell Signaling Technologies | 2012 | 1:1000 |
| Antibody | HRP-anti-Rabbit IgG (Goat polyclonal) | Cell Signaling Technologies | 7074 | 1:5000 |
| Antibody | HRP-anti-Mouse IgG (Goat polyclonal) | Cell Signaling Technologies | 7076 | 1:5000 |
| Transfected construct (human) | MYC | Vector Biolabs | 1285 | adenovirus to express MYC |
| Transfected construct (human) | YFP | *Oldham et al., 2015* | | adenovirus control to express YFP |
| Chemical compound, drug | [1,2–13 C2]-glucose | Cambridge Isotope Labs | CLM-504-PK | |
| Chemical compound, drug | [U-13C6]-glucose | Cambridge Isotope Labs | CLM-1396-PK | |
| Chemical compound, drug | [U-13C5]-glutamine | Cambridge Isotope Labs | CLM-1822-H-PK | |
| Chemical compound, drug | [U-13C3]-lactate | Sigma | 485926 | |
| Chemical compound, drug | BAY | Cayman | 15297 | Molidustat (BAY-85–3934); 10 µM in DMSO |
| Commercial assay or kit | Glucose colorimetric assay kit | Cayman | 10009582 | |
| Commercial assay or kit | L-Lactate assay kit | Cayman | 700510 | |
| Commercial assay or kit | Pyruvate assay kit | Cayman | 700470 | |
| Commercial assay or kit | NADP/NADPH-Glo Assay | Promega | G9081 | |
| Cell line (human) | LFs | Lonza | CC-2512 | Normal human lung fibroblasts |
| Cell line (human) | PASMCs | Lonza | CC-2581 | Pulmonary artery smooth muscle cells |
| Sequence-based reagent | ACTB | Life Technologies | Hs03023943_g1 | qPCR probe |
| Sequence-based reagent | GLUT1 | Life Technologies | Hs00892681_m1 | qPCR probe |
| Sequence-based reagent | LDHA | Life Technologies | Hs00855332_g1 | qPCR probe |
| Sequence-based reagent | CDKN1A | Life Technologies | Hs00355782_m1 | qPCR probe |
| Transfected construct (human) | siMYC | Dharmacon | L-003282-02-0005 | ON-TARGETplus siRNA |
| Transfected construct (human) | siPHD2 | Dharmacon | L-004276-00-0005 | ON-TARGETplus siRNA |
| Transfected construct (human) | siCTL | Dharmacon | D-001810-10-05 | ON-TARGETplus non-targeting control pool |

## Cell culture

Primary normal human lung fibroblasts (LFs) were purchased from Lonza (CC-2512) and cultured in FGM-2 (Lonza CC-3132). Cells from three donors were used in these studies: #33652 (56 y.o., male), #29132 (19 y.o., female), and #41684 (37 y.o., male). Passages 3–6 were used for experiments. Primary human pulmonary artery smooth muscle cells were purchased from Lonza (CC-2581) and cultured in SmGM-2 (Lonza CC-3182). Cells from three donors were used in these studies: #30020 (64 y.o., male), #27662 (35 y.o., male), #26698 (51 y.o., male), #19828 (51 y.o., male) and #45518 (56 y.o., female). Passages 4–7 were used for experiments. Cell authentication was performed by the vendor. Cells were maintained in a standard tissue culture incubator in 5% $CO_2$ at 37 °C.

## Metabolic flux protocol

For extracellular flux measurements, cells were seeded in either standard growth medium or MCDB131 medium lacking glucose, glutamine, and phenol red (genDEPOT) which was supplemented with 2%

dialyzed fetal bovine serum (Mediatech) and naturally labeled glucose and glutamine ('light' labeling medium). For LFs, glucose was supplemented at 8 mM and glutamine was supplemented at 1 mM. For PASMCs, glucose was supplemented at 5.55 mM and glutamine was supplemented at 10 mM. These concentrations match the concentrations of these substrates determined in standard growth medium. Preliminary experiments were performed to identify the optimal cell seeding density, exponential growth phase, and labeling duration consistent with metabolic and isotopic steady state. On Day −1, 25,000 cells were seeded in a 35 mm dish in 'light' labeling medium. Hypoxia-treated cells were transferred to a tissue culture glovebox set at 0.5% oxygen and 5% $CO_2$ (Coy Lab Products). Medium was supplemented with DMSO 0.1% or BAY (10 μM) for these treatment conditions. On Day 0, cells were washed with PBS and the medium was changed to either 'light' labeling medium for flux measurements or 'heavy' labeling medium containing [1,2-$^{13}C_2$]-glucose, [U-$^{13}C_6$]-glucose, [U-$^{13}C_5$]-glutamine, or [U-$^{13}C_2$]-lactate for tracer experiments. For LFs, samples were collected on Day 0 and every 24 hr for 72 hr. For PASMCs, samples were collected on Day 0 and every 12 hr for 48 hr. Medium and cell lysates were collected at each time point for intra- and extracellular metabolite measurements and total DNA quantification. Dishes without cells were weighed daily to correct for evaporative medium losses and to empirically determine degradation and accumulation rates of metabolites. Medium samples and cell lysates for DNA measurement were stored at −80 °C until analysis. Each individual experiment included triplicate wells for each treatment and time point, and each experiment was repeated 4–8 times.

## Cell count

Direct cell counts of trypsinized cell suspensions in PBS were obtained following staining with propidium iodide and acridine orange using a LUNA-FL fluorescence cell counter (Logos Biosystems). Indirect cell counts for flux measurements were interpolated from total DNA quantified using the Quant-iT PicoGreen dsDNA Assay Kit (Thermo). Cells were washed once with two volumes of PBS, lysed with Tris-EDTA buffer containing 2% Triton X-100, and collected by scraping. Total DNA in 10 μL of lysate was determined by adding 100 μL of 1 X PicoGreen dye in Tris-EDTA buffer and interpolating the fluorescence intensity with a standard curve generated using the $\lambda$ DNA standard. Cell counts were interpolated from a standard curve of DNA obtained from known cell numbers seeded in basal medium (*Figure 12A*). No difference in total cellular DNA was identified between normoxia and hypoxia cultures (*Figure 12B–C*).

## Immunoblots

Cells were washed with one volume of PBS and collected by scraping in PBS. Cell suspensions were centrifuged at 5000×g for 5 min at 4 °C. Pellets were lysed in buffer containing Tris 10 mM, pH 7.4, NaCl 150 mM, EDTA 1 mM, EGTA 1 mM, Triton X-100 1% v/v, NP-40 0.5% v/v, and Halt Protease Inhibitor Cocktail (Thermo). Protein concentrations were determined by BCA Protein Assay (Thermo).

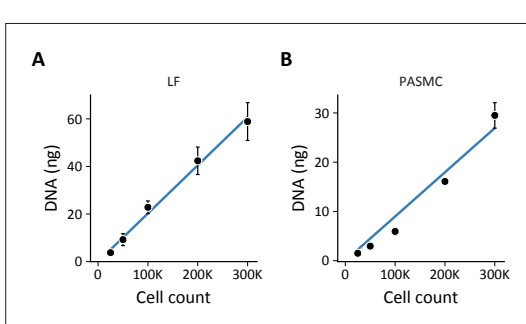

Lysates were normalized for protein concentration and subjected to SDS-PAGE separation on stain-free tris-glycine gels (Bio-Rad), cross-linked and imaged with the Chemidoc system (Bio-Rad), transferred to PVDF membranes with the Trans-Blot Turbo transfer system (Bio-Rad), imaged, blocked in 5% blocking buffer (Bio-Rad), blotted in primary and secondary antibodies, and developed using WesternBright ECL (Advansta). Band signal intensity was normalized to total protein per lane as determined from the stain-free gel or membrane images.

## RT-qPCR

Total RNA was isolated from cells with the RNeasy Mini Kit (Qiagen). cDNA was synthesized from 0.25 to 1.00 ng RNA with the High Capacity cDNA Reverse Transcription Kit (Applied Biosystems).

**Figure 12.** Total DNA as a surrogate for cell count. (**A**) Standard curve of lung fibroblast (LF) cell count *v.* total DNA by PicoGreen measurement used to interpolate cell numbers from DNA measurements. Data are mean ± SEM of three biological replicates. (**B**) Standard curve of PASMC cell count *v.* total DNA as in (**A**).

RT-qPCR analysis was performed with an Applied Biosystems 7500 Fast Real Time PCR System with TaqMan Universal PCR Master Mix and pre-designed TaqMan gene expression assays (Life Technologies). Relative expression levels were calculated using the comparative cycle threshold method referenced to *ACTB*.

### Glucose assay
Medium samples were diluted 10-fold in PBS. Glucose concentration was determined using the Glucose Colorimetric Assay Kit (Cayman) according to the manufacturer's protocol. Standards were prepared in PBS.

### Lactate assay
Medium samples were diluted 10-fold in PBS. Glucose concentration was determined using the L-Lactate Assay Kit (Cayman). Medium samples did not require deproteinization, otherwise the samples were analyzed according to the manufacturer's protocol. Standards were prepared in PBS.

### Pyruvate assay
Pyruvate was measured using either an enzymatic assay (most samples) or an HPLC-based assay (medium from 0.2% oxygen experiments). For the enzymatic assay, medium samples were diluted 20-fold in PBS. Pyruvate concentration was determined using the Pyruvate Assay Kit (Cayman). Medium samples did not require deproteinization, otherwise the samples were analyzed according to the manufacturer's protocol. Standards were prepared in PBS. For the HPLC assay, 2-oxovaleric acid was added to medium samples as an internal standard. Samples were subsequently deproteinized with 2 volumes of ice-cold acetone. Supernatants were evaporated to <50% of the starting volume at 43 °C in a SpeedVac concentrator (Thermo Savant) and reconstituted to the starting volume with HPLC-grade water prior to derivatization. Samples were derivatized 1:1 by volume with *o*-phenylene-diamine (25 mM in 2 M HCl) for 30 min at 80 °C. Derivatized pyruvate was separated with a Poroshell HPH C-18 column (2.1×100 mm, 2.7 μm) on an Infinity II high-performance liquid chromatography system with fluorescence detection of OPD-derivatized α-keto acids as described previously (*Guarino et al., 2019*).

### Amino acid assay
Medium amino acid concentrations were determined following the addition of norvaline and sarcosine internal standards and deproteinization with 2 volumes of ice-cold acetone. Supernatants were evaporated to <50% of the starting volume at 43 °C in a SpeedVac concentrator (Thermo Savant) and reconstituted to the starting volume with HPLC-grade water prior to analysis. Amino acids in deproteinized medium were derivatized with *o*-phthalaldehyde (OPA) and 9-fluorenylmethylchloroformate (FMOC) immediately prior to separation with a Poroshell HPH-C18 column (4.6×100 mm, 2.7 μm) on an Infinity II high-performance liquid chromatography system with ultraviolet and fluorescence detection of OPA- and FMOC-derivatized amino acids, respectively, according to the manufacturer's protocol (Agilent; *Long, 2017*).

### RNA interference
Approximately 1.25 M LFs were reverse transfected in 6 cm dishes with 40 pmol siRNA or non-targeting siCTL pools (Dharmacon) and 20 μL RNAiMAX (Thermo) in 500 μL OptiMEM (Thermo). After 24 hr, cells were collected by trypsinization and re-seeded as described in *Metabolic flux protocol* above for growth rate and lactate efflux measurements.

### MYC overexpression
LFs were seeded at 25,000 cells per 35 mm dish on Day –2. On Day –1, cells were transduced with adenovirus for MYC (Vector Biolabs) or YFP overexpression (*Oldham et al., 2015*). After 24 hr, the medium was changed and samples were collected as described in *Metabolic flux protocol* above.

### Flux calculations
The growth rate (μ) and flux (*v*) for each measured metabolite were defined as follows *Murphy and Young, 2013*:

$$\frac{dX}{dt} = \mu X \tag{1}$$

$$\frac{dM}{dt} = -kM + vX \tag{2}$$

where $X$ is the cell density, $k$ is the first-order degradation or accumulation rate, and $M$ is the mass of the metabolite. These equations are solved as follows:

$$X = X_0 e^{\mu t} \tag{3}$$

$$Me^{kt} = \frac{vX_0}{\mu + k}\left(e^{(\mu+k)t} - 1\right) + M_0 \tag{4}$$

Growth rate ($\mu$) and cell count at time 0 ($X_0$) were determined by robust linear modeling of the logarithm of cell count as a function of time ($t$). Metabolite mass was calculated from the measured metabolite concentrations and predicted well volume accounting for evaporative losses (*Figure 1— figure supplement 1E*). First-order degradation and accumulation rates were obtained from robust linear modeling of metabolite mass *v*. time in unconditioned culture medium. Rates that significantly differed from 0 by Student's *t*-test were incorporated into the flux calculations. Final fluxes were obtained by robust linear modeling of $Me^{kt}$ *versus* $\left(e^{(\mu+k)t} - 1\right)$ to determine the slope from which $v$ was calculated using *Equation 4*.

## Metabolomics
### Metabolite extraction
Intracellular metabolites were obtained after washing cells with 2 volumes of ice-cold PBS and floating on liquid nitrogen. Plates were stored at –80 °C until extraction. Metabolites were extracted with 1 mL 80% MeOH pre-cooled to –80 °C containing 10 nmol [D$_8$]-valine as an internal standard (Cambridge Isotope Labs). Insoluble material was removed by centrifugation at 21,000×*g* for 15 min at 4 °C. The supernatant was evaporated to dryness at 42 °C using a SpeedVac concentrator (Thermo Savant). Samples were resuspended in 35 µL LC-MS-grade water prior to analysis.

### Acquisition parameters
LC-MS analysis was performed on a Vanquish ultra-high-performance liquid chromatography system coupled to a Q Exactive orbitrap mass spectrometer by a HESI-II electrospray ionization probe (Thermo). External mass calibration was performed weekly. Metabolite samples (2.5 µL) were separated using a ZIC-pHILIC stationary phase (2.1×150 mm, 5 µm; Merck). The autosampler temperature was 4 °C and the column compartment was maintained at 25 °C. Mobile phase A was 20 mM ammonium carbonate and 0.1% ammonium hydroxide. Mobile phase B was acetonitrile. The flow rate was 0.1 mL/min. Solvent was introduced to the mass spectrometer *via* electrospray ionization with the following source parameters: sheath gas 40, auxiliary gas 15, sweep gas 1, spray voltage +3.0 kV for positive mode and –3.1 kV for negative mode, capillary temperature 275 °C, S-lens RF level 40, and probe temperature 350 °C. Data were acquired and peaks integrated using TraceFinder 4.1 (Thermo).

### Stable isotope quantification
All metabolites except fructose 2,6-bisphosphate (FBP) and 3-phosphoglycerate (3 PG) were measured using the following mobile phase gradient: 0 min, 80% B; 5 min, 80% B; 30 min, 20% B; 31 min, 80% B; 42 min, 80% B. The mass spectrometer was operated in selected ion monitoring mode with an m/z window width of 9.0 centered 1.003355-times half the number of carbon atoms in the target metabolite. The resolution was set at 70,000 and AGC target was 1×10$^5$ ions. Peak areas were corrected for quadrupole bias as in *Kim et al., 2015*. Mass isotope distributions for FBP and 3 PG were calculated from full scan chromatograms as described below. Raw mass isotopomer distributions were corrected for natural isotope abundance using a custom R package (mzrtools; https://github.com/wmoldham/ mzrtools; *Oldham, 2020*) employing the method of *Fernandez et al., 1996*.

## Metabolomic profiling

For metabolomic profiling and quantification of isotopic enrichment for FBP and 3 PG, the following mobile phase gradient was used: 0 min, 80% B; 20 min, 20% B; 20.5 min, 80% B; 28 min, 80% B; 42 min, 80% B. The mass spectrometer was operated in polarity switching full scan mode from 70 to 1000 m/$z$. Resolution was set to 70,000 and the AGC target was 1×10$^6$ ions. Peak identifications were based on an in-house library of authentic metabolite standards previously analyzed utilizing this method. For metabolomics studies, pooled quality control (QC) samples were injected at the beginning, end, and between every four samples of the run. Raw peak areas for each metabolite were corrected for instrument drift using a cubic spline model of QC peak areas. Low-quality features were removed on the basis of a relative standard deviation greater than 0.2 in the QC samples and a dispersion ratio greater than 0.4 (*Broadhurst et al., 2018*). Missing values were imputed using random forest. Sample peak areas were normalized using probabilistic quotient normalization (*Dieterle et al., 2006*). Differentially regulated metabolites were identified using limma (*Ritchie et al., 2015*). Metabolite set enrichment analysis was performed using the fgsea package (*Korotkevich et al., 2021*) with KEGG metabolite pathways (*Kanehisa and Goto, 2000*).

## Biomass determination

The dry weight of LFs was determined to be 493 pg/cell. The dry weight of PASMCs was determined to be 396 pg/cell. These values were estimated by washing 3×10$^6$ cells twice in PBS and thrice in ice-cold acetone prior to drying overnight in a SpeedVac. The composition of the dry cell mass was estimated from the literature (*Quek et al., 2010*; *Sheikh et al., 2005*), and stoichiometric coefficients were determined as described (*Murphy and Young, 2013*; *Zamorano et al., 2010*).

## Metabolic flux analysis

Metabolic flux analysis was performed using the elementary metabolite unit-based software package INCA (*Young, 2014*). Inputs to the model include the chemical reactions and atom transitions of central carbon metabolism, extracellular fluxes, the identity and composition of $^{13}$C-labeled tracers, and the MIDs of labeled intracellular metabolites. The metabolic network was adapted from previously published networks (*Murphy and Young, 2013*; *Vacanti et al., 2014*) and comprises 48 reactions representing glycolysis, the pentose phosphate pathway, the tricarboxylic acid cycle, anaplerotic pathways, serine metabolism, and biomass synthesis. The network includes seven extracellular substrates (aspartate, cystine, glucose, glutamine, glycine, pyruvate, serine) and five metabolic products (alanine, biomass, glutamate, lactate, lipid). Models were fit using three $^{13}$C-labeled tracers, [1,2-$^{13}$C$_2$] glucose, [U-$^{13}$C$_6$] glucose, and [U-$^{13}$C$_5$] glutamine. The MIDs of twelve metabolites (2-oxoglutarate, 3-phosphoglycerate, alanine, aspartate, citrate, fructose bisphosphate, glutamate, glutamine, lactate, malate, pyruvate, serine) were used to constrain intracellular fluxes. The following assumptions were made:

1. Metabolism was at steady state.
2. Labeled CO$_2$ produced during decarboxylation reactions left the system and did not re-incorporate during carboxylation reactions.
3. Protein turnover occurred at a negligible rate compared to glucose and glutamine consumption.
4. Acetyl-CoA and pyruvate existed in cytosolic and mitochondrial pools. Aspartate, fumarate, oxaloacetate, and malate were allowed to exchange freely between the compartments.
5. The per cell biomass requirements of LFs and PASMCs were similar to published estimates from other cell types (*Quek et al., 2010*; *Sheikh et al., 2005*).
6. Succinate and fumarate are symmetric molecules that have interchangeable orientations when metabolized by TCA cycle enzymes.

Flux estimation was repeated a minimum of 50 times from random initial values. Results were subjected to a $\chi^2$ statistical test to assess goodness-of-fit. Accurate 95% confidence intervals were computed for estimated parameters by evaluating the sensitivity of the sum-of-square residuals to parameter variations (*Antoniewicz et al., 2006*; *Murphy and Young, 2013*).

## NAD(H) and NADP(H) assays

Cellular NAD$^+$ and NADH were measured using an enzymatic fluorometric cycling assay based on the reduction of NAD$^+$ to NADH by alcohol dehydrogenase (ADH) and subsequent electron transfer to

generate the fluorescent molecule resorufin (*Oldham et al., 2015*). Briefly, cells were washed twice with one volume PBS. Pyridine nucleotides were extracted on ice with buffer containing 50% by volume PBS and 50% lysis buffer (100 mM sodium carbonate, 20 mM sodium bicarbonate, 10 mM nicotinamide, 0.05% by volume Triton-X-100, 1% by mass dodecyltrimethylammonium bromide) and collected by scraping. Extracts were divided equally and 0.5 volume of 0.4 N HCl was added to one sample. Both extracts were heated at 65 °C for 15 min to degrade selectively either the oxidized (buffer) or reduced (HCl) nucleotides. The reaction was cooled on ice and quenched by adding 0.5 M Tris-OH to the acid-treated samples or 0.2 N HCl plus 0.25 M Tris-OH to the buffer samples. Samples were then diluted in reaction buffer (50 mM EDTA and 10 mM Tris, pH 7.06). Cell debris was pelleted by centrifugation, and 50 μL was incubated for 2 h with 100 μL reaction buffer containing 0.6 M EtOH, 0.5 mM phenazine methosulfate, 0.05 mM resazurin, and 0.1 mg/mL ADH. Fluorescence intensities were measured with a Spectramax Gemini XPS (Molecular Devices) with excitation 540 nm, emission 588 nm, and 550 nm excitation cut-off filter. Sample intensities were compared to a standard curve generated from known concentrations of NADH. The ratio of fluorescence in buffer-extracted to acid-extracted samples corresponds to the NADH/NAD$^+$ ratio. Absolute NADH and NAD$^+$ were normalized to estimated cell counts from total DNA quantification as described above.

Samples for NADP$^+$ and NADPH quantification were prepared as described above and analyzed using the luminescence-based NADP/NADPH-Glo assay (Promega).

## RNA-seq

RNA was collected from LFs treated for three days ± hypoxia ± BAY as described above. Four biological replicates were analyzed. Library construction and sequencing was performed by BGI Genomics using 100 bp paired end analysis and a read depth of 50 M reads per sample. Sequences were deposited in the NIH SRA (PRJNA721596). Sequences were mapped to the human GRCh38 primary assembly and counts summarized using Rsubread (*Liao et al., 2019*). This data is available from the Oldham Lab GitHub repository (https://github.com/oldhamlab/rnaseq.lf.hypoxia.molidustat; *Oldham, 2021* ). Differentially expressed transcripts were identified using DESeq2 (*Love et al., 2014*). Gene set enrichment and transcription factor enrichment was performed using the fgsea and DoRothEA R packages, respectively (*Garcia-Alonso et al., 2019*; *Korotkevich et al., 2021*).

## Quantification and statistical analysis

The raw data and annotated analysis code necessary to reproduce this manuscript are contained in an R package research compendium available from the Oldham Lab GitHub repository (https://github.com/oldhamlab/Copeland.2023.hypoxia.flux; *Oldham, 2023*). Data analysis, statistical comparisons, and visualization were performed in R (*R Development Core Team, 2023*). Experiments included technical and biological replicates as noted above. The number of biological replicates (n) is indicated in the figure legends. Summary data show the mean ± SEM. Outliers were identified using twice the median absolute deviation as a cutoff threshold. Comparisons were performed using linear mixed-effects models with oxygen, treatment, and their interaction as fixed effects and biological replicate as a random effect. Significant differences in estimated marginal means were identified by comparisons to the multivariate *t* distribution. Metabolomics and RNA-seq data were analyzed as described above. Probability values less than 0.05 were considered statistically significant.

## Acknowledgements

This work was supported by grants from the NIH (K08HL128802), American Lung Association, Pulmonary Hypertension Association, and the American Thoracic Society Foundation to W.M.O and from the NIH (U01HG007690, U54HL119145, R01HL155107, R01HL155096) and the American Heart Association (D700382, CV-19) to J.L.

## Additional information

### Funding

| Funder | Grant reference number | Author |
|---|---|---|
| National Institutes of Health | K08HL128802 | William M Oldham |
| American Lung Association | RG-415134 | William M Oldham |
| Pulmonary Hypertension Association | | William M Oldham |
| American Thoracic Society | | William M Oldham |
| National Institutes of Health | U01HG007690 | Joseph Loscalzo |
| National Institutes of Health | U54HL119145 | Joseph Loscalzo |
| National Institutes of Health | R01HL155107 | Joseph Loscalzo |
| National Institutes of Health | R01HL155096 | Joseph Loscalzo |
| American Heart Association | D700382 | Joseph Loscalzo |
| American Heart Association | CV-19 | Joseph Loscalzo |

The funders had no role in study design, data collection and interpretation, or the decision to submit the work for publication.

### Author contributions

Courtney A Copeland, David Ziehr, Sarah McGarrity, Formal analysis, Investigation, Writing – review and editing; Benjamin A Olenchock, Formal analysis, Investigation, Methodology, Writing – review and editing; Kevin Leahy, Investigation, Writing – review and editing; Jamey D Young, Software, Methodology, Writing – review and editing; Joseph Loscalzo, Resources, Supervision, Funding acquisition, Writing – review and editing; William M Oldham, Conceptualization, Resources, Data curation, Software, Formal analysis, Supervision, Funding acquisition, Investigation, Visualization, Methodology, Writing – original draft, Project administration, Writing – review and editing

### Author ORCIDs

Joseph Loscalzo (iD) http://orcid.org/0000-0002-1153-8047
William M Oldham (iD) http://orcid.org/0000-0003-3029-4866

### Decision letter and Author response

Decision letter https://doi.org/10.7554/eLife.82597.sa1
Author response https://doi.org/10.7554/eLife.82597.sa2

---

## Additional files

### Supplementary files

• MDAR checklist

### Data availability

RNA sequencing data were deposited in the NIH SRA under the accession code PRJNA721596.The raw data and annotated analysis code necessary to reproduce this manuscript are contained in an R package research compendium available from the Oldham Lab GitHub repository (https://github.com/oldhamlab/Copeland.2023.hypoxia.flux, copy archived at *Oldham, 2023*).

The following dataset was generated:

| Author(s) | Year | Dataset title | Dataset URL | Database and Identifier |
|---|---|---|---|---|
| Oldham WM | 2023 | MYC overrides HIF-1α to regulate proliferating primary cell metabolism in hypoxia | https://www.ncbi.nlm.nih.gov/bioproject/?term=PRJNA721596 | NCBI BioProject, PRJNA721596 |

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
