## [Editor Report]

The manuscript by Copeland and colleagues describes the impact of HIF1a, MYC and metabolism in pulmonary lung fibroblast and pulmonary artery smooth muscle cell phenotype, which is highly relevant to pulmonary vascular disease. The work includes metabolic flux assays of cultured cells, using a combination of metabolite concentration assessments, stable isotope-labeled substrates coupled with mass spectrometry, mathematical modeling, and cell proliferation analysis. Overall the findings are that there is an unexpected drop in lactate production in hypoxia and with HIF augmentation. These studies will add to the field's understanding of the role of HIF and cellular metabolism in pulmonary hypertension.

---

## [Decision Letter]

**Decision letter after peer review:**

Thank you for submitting your article "MYC overrides HIF to regulate proliferating primary cell metabolism in hypoxia" for consideration by *eLife*. Your article has been reviewed by 3 peer reviewers, and the evaluation has been overseen by a Reviewing Editor and Paul Noble as the Senior Editor. The following individual involved in the review of your submission has agreed to reveal their identity: Brian Graham (Reviewer #1).

The reviewers have discussed their reviews with one another, and the Reviewing Editor asks you to address the comments of the three thoughtful reviews below.

*Reviewer #1 (Recommendations for the authors):*

1. In general the manuscript and figures are relatively dense and sometimes hard to follow through the various permutations of hypoxia vs normoxia, BAY vs DMSO, LF vs PASMCs, different substrates, and MYC KD or OE. I think the manuscript would benefit from streamlining the text and figures without exploring all the minute details, as well as a summary cartoon explaining how the different pathways relate to metabolic features and cell phenotype.

2. Title: suggest revising to HIF1a specifically as other isoforms are not explored.

3. Cells: use of primary cultured cells (even if not immortalized) is a limitation that should be explored. The media contains FBS and growth supplements to promote ex vivo growth, that may not be replicated by in vivo conditions. What passage were the experiments performed at? Repeated passage will progressively select for more robustly growing cells, which again are less likely to recapitulate the heterogenous population present in vivo.

4. The finding of decreased lactate efflux in hypoxia needs to be more developed to further increase confidence in this finding. Suggestions: assessment of ECAR, and 13-C labeled lactate in the supernatant or pellet from 13C-labeled glucose substrate. Also, can you add 1 sentence in the results second paragraph briefly summarizing how the data in Figure 1 were generated (versus the more broad summary of many different approaches in the first paragraph)? Related: it is noted (line 200) that 10% of cytoplasmic pyruvate enters the TCA cycle, with the balance converted to lactate. Is this at baseline? If so, practically is there room for an additional increase in pyruvate to be converted to lactate with hypoxia or BAY treatment?

5. There is an increase in labeled lactate uptake noted (Figure 4); the data in panel C don't appear to support an increase in lactate oxidation-can you clarify this? Also, does the modeling used in generating the data in the prior figures account for the possibility of lactate recirculation? Given lactate as a substrate seems to be a substantial contribution to cell metabolism, this is an important consideration.

6. There is an exploration of NADH and NAD discussed in lines 292-294 but did you also look at NADPH? This would more naturally follow the data on increased PPP flux which was discussed previously (lines 203-204).

7. There is evidence that MYC can directly impact HIF1a, such as post-transcriptionally (see PMID: 22186139). In the MYC knockdown/overexpression experiments, can you assess HIF1a protein and some HIF1a targets as you did in earlier figures? I think this will help clarify the role of MYC vis-à-vis HIF1a in these cells.

*Reviewer #2 (Recommendations for the authors):*

1. The authors should test whether hypoxia increases dependence on glucose or glutamine (for residual cell growth or viability) in hypoxia. They comment that in LFs, hypoxia reduces glutamine utilization and contrast this with some published papers, including the Wise et al. 2011 paper. But an important finding of that paper was increased reliance on glutamine for growth. It would be helpful to test that here.

2. The BAY compound "uncouples cell proliferation and metabolic flux" (lines 222-223). This is true based on the data, but it is not clear that the mechanism is through PHD inhibition. This drug is the only approach to induce normoxic HIF-a stabilization in the paper. It would increase confidence in the findings of the BAY compound if its effects could be reproduced with genetic silencing of PHD.

3. The metabolic characterization is solid and thorough, but the authors present some fairly surprising data and do not explain how these occur. For example, it is unclear how MYC is induced under hypoxia. It is also unclear why MYC's metabolic effects in hypoxia are the opposite of what it does in normoxia. LDHA is a classical MYC target, so why MYC suppresses lactate secretion in these cells is mysterious. Given that the major conclusion of the paper is that hypoxia brings about unexpected metabolic effects in non-transformed cells, I feel there needs to be more clarity about how these effects occur.

*Reviewer #3 (Recommendations for the authors):*

I felt the strengths of this paper are substantial, and the impact of the findings will set the stage for further studies contrasting the molecular and metabolic effects of hypoxia in primary vs. transformed cells.

1. Since the paper is predicated on the comparison of primary vs. transformed cells, it would be appropriate if the authors could repeat some key experiments with transformed cells to show the side-by-side difference of primary vs. transformed cells in MYC-dependent reduction of glycolytic activity.

2. Is this MYC-dependent reduction of glycolytic activity also similarly seen in primary cells with fewer mitochondria than LFs and PASMCs (such as endothelium)?

3. Can the authors confirm that MYC is up-regulated in hypoxic primary tissue in normal physiologic conditions in vivo?

4. The presentation of results can be a bit difficult to follow, particularly since the authors often will put in commentary and references of known functions of various metabolites in the actual Results. I suggest that they save the commentary for the Discussion and simply report data in the Results.

---

## [Author Response]

Reviewer #1 (Recommendations for the authors):1. In general the manuscript and figures are relatively dense and sometimes hard to follow through the various permutations of hypoxia vs normoxia, BAY vs DMSO, LF vs PASMCs, different substrates, and MYC KD or OE. I think the manuscript would benefit from streamlining the text and figures without exploring all the minute details, as well as a summary cartoon explaining how the different pathways relate to metabolic features and cell phenotype.

We appreciate this comment and have reviewed the manuscript carefully with an eye toward streamlining the data presentation. We made some progress but were confronted with the reality that the manuscript includes a lot of data. Where we focus on more minute details, we have attempted to provide context for the observations by referencing prior investigations of hypoxic cell metabolism using similar methods, which we believe will be of interest to the metabolism research community. With the flexibility afforded by *eLife*, we have divided several of the figures into groups of smaller panels to focus on an individual message. We have also redrawn Figure 11C to more clearly illustrate the relationships between HIF and MYC following normoxic and hypoxic prolyl hydroxylase stabilization.

2. Title: suggest revising to HIF1a specifically as other isoforms are not explored.

We agree and have changed the title accordingly.

3. Cells: use of primary cultured cells (even if not immortalized) is a limitation that should be explored. The media contains FBS and growth supplements to promote ex vivo growth, that may not be replicated by in vivo conditions. What passage were the experiments performed at? Repeated passage will progressively select for more robustly growing cells, which again are less likely to recapitulate the heterogenous population present in vivo.

We have now included this information in the Methods. LFs were used between passages 3-6 and PASMCs were used between passages 4-7. We did not observe any passage-dependent effects across experiments, which were typically performed in multiple donors across different passages. We certainly agree that there are limitations to in vitro cell cultures as a model system of cell biology in vivo and are looking forward to investigating these regulatory pathways in relevant animal models going forward. Nonetheless, we think our finding that HIF-1α stabilization has differential effects on cell metabolism in normoxia and hypoxia is quite a novel and unexpected result even in comparison to prior tissue culture-based experiments.

4. The finding of decreased lactate efflux in hypoxia needs to be more developed to further increase confidence in this finding. Suggestions: assessment of ECAR, and 13-C labeled lactate in the supernatant or pellet from 13C-labeled glucose substrate. Also, can you add 1 sentence in the results second paragraph briefly summarizing how the data in Figure 1 were generated (versus the more broad summary of many different approaches in the first paragraph)? Related: it is noted (line 200) that 10% of cytoplasmic pyruvate enters the TCA cycle, with the balance converted to lactate. Is this at baseline? If so, practically is there room for an additional increase in pyruvate to be converted to lactate with hypoxia or BAY treatment?

As mentioned above and suggested by Reviewer 1, we now include lactate efflux measurements based on the metabolism of [U-^13^C_6_]-glucose to [U-^13^C_3_]-lactate as measured by stable isotope resolved liquid chromatography-mass spectrometry analysis of extracellular lactate (Figure 1 —figure supplement 2). This is the most direct measurement of glycolysis and lactate fermentation and would be considered the gold standard for these analyses (*i.e.*, tracing of a particular labeled substrate to a particular labeled product). Consistent with the data previously obtained using lactate concentrations determined by enzyme assay, we observe hypoxia-dependent decreases in lactate efflux in lung fibroblasts.

In support of this finding, we can use the data presented previously to calculate intracellular [U-^13^C_3_]-lactate. In the graph in Author response image 1, the M3 lactate isotope peak intensities were divided by cell count across the conditions studied. Here, hypoxia decreases intracellular [U-^13^C_3_]-lactate.

**Author response image 1. sa2fig1:** 

Extracellular acidification rate measurements (ECAR) from the Seahorse bioanalyzer are the most indirect measurement of lactate efflux. These data are confounded by non-glycolytic acidification, including oxidative phosphorylation. For these reasons, we did not pursue these measurements.Regarding the flux calculations themselves, we now refer the reader to the Materials and methods section where these calculations were described in detail. Essentially, fluxes are calculated from the slope of the best-fit line of metabolite mass (*y* axis) and an exponential term incorporating cell growth rate (Equation 4).

We have clarified that 10% of cytoplasmic pyruvate enters the TCA at baseline by adding “In normoxia” at the beginning of this sentence. It’s important to distinguish between the fractional distribution of metabolism and the flux through the metabolic pathways. Metabolite fluxes can change dramatically without requiring a change in the fractional disposition of substrates (*i.e.*, glycolysis could increase 2-3-fold while still sending 10% of pyruvate to the TCA cycle). Indeed, the reason we performed the initial BAY experiments was to ensure that HIF stabilization would augment glycolysis and lactate efflux, which it did by ~ 1.5-fold. Moreover, both hypoxia and BAY treatment increased the number of glycolytic proteins, suggesting that the overall capacity for glycolysis should be increased.

5. There is an increase in labeled lactate uptake noted (Figure 4); the data in panel C don't appear to support an increase in lactate oxidation-can you clarify this? Also, does the modeling used in generating the data in the prior figures account for the possibility of lactate recirculation? Given lactate as a substrate seems to be a substantial contribution to cell metabolism, this is an important consideration.

We have now added α-ketoglutarate and malate to the data now presented in Figure 7C. These data demonstrate increased oxidation of lactate to pyruvate based on increased M3 pyruvate as well as increased oxidation in the TCA cycle as demonstrated by increased M2 AKG and M3 malate.

The modeling does incorporate reversible reactions, and lactate transport was modeled as a reversible reaction. To emphasize this point, we have added a sentence clarifying the definition of “exchange” fluxes.

6. There is an exploration of NADH and NAD discussed in lines 292-294 but did you also look at NADPH? This would more naturally follow the data on increased PPP flux which was discussed previously (lines 203-204).

These data are now included as Figure 9 —figure supplement 2. We observed hypoxia-dependent decreases in NADPH/NADP ratio independent of BAY treatment.

7. There is evidence that MYC can directly impact HIF1a, such as post-transcriptionally (see PMID: 22186139). In the MYC knockdown/overexpression experiments, can you assess HIF1a protein and some HIF1a targets as you did in earlier figures? I think this will help clarify the role of MYC vis-à-vis HIF1a in these cells.

We have now included these data as Figure 11 —figure supplement 1. We did not observe any significant impact of siMYC treatment on HIF target gene expression in hypoxia. We found that MYC overexpression increased HIF protein and target gene expression levels.

Reviewer #2 (Recommendations for the authors):1. The authors should test whether hypoxia increases dependence on glucose or glutamine (for residual cell growth or viability) in hypoxia. They comment that in LFs, hypoxia reduces glutamine utilization and contrast this with some published papers, including the Wise et al. 2011 paper. But an important finding of that paper was increased reliance on glutamine for growth. It would be helpful to test that here.

We have included these data as Figure 3. In LFs, we find that cell growth is decreased by either glucose or glutamine deprivation. Cell growth rate decreases further with glucose deprivation in hypoxia, but hypoxia did not affect cell growth in the absence of glutamine. In PASMCs, we did not observe significant differences in cell growth rates in the absence of either glucose or glutamine, although glucose deficiency tended to be associated with decreased growth. Interestingly, growth rates were unaffected by the absence of glutamine despite increased glutamine uptake that we observed in hypoxia in our flux models. Together, these data suggest a degree of metabolic flexibility in PASMCs such that they can compensate for glutamine deficiency. Characterizing these changes further would require measuring substrate fluxes in the presence and absence of glutamine, which we considered to be beyond the scope of the present work.

2. The BAY compound "uncouples cell proliferation and metabolic flux" (lines 222-223). This is true based on the data, but it is not clear that the mechanism is through PHD inhibition. This drug is the only approach to induce normoxic HIF-a stabilization in the paper. It would increase confidence in the findings of the BAY compound if its effects could be reproduced with genetic silencing of PHD.

We have now included siPHD2 in analyses similar to BAY treatment. These data support our findings with BAY. We show that siPHD2 stabilizes HIF-1α and target gene expression in normoxia. Additionally, hypoxia attenuates siPHD2-dependent increases in lactate efflux, similar to BAY treatment (Figure 8 —figure supplement 1). These effects are likely mediated by MYC based on similarities in CDKN1A expression (Figure 10 —figure supplement 2), and we observe similar effects of hypoxia on MYC expression in siPHD2-treated cells (Figure 11 —figure supplement 1). Together, these data bolster our conclusions from experiments on BAY-treated cells.

3. The metabolic characterization is solid and thorough, but the authors present some fairly surprising data and do not explain how these occur. For example, it is unclear how MYC is induced under hypoxia. It is also unclear why MYC's metabolic effects in hypoxia are the opposite of what it does in normoxia. LDHA is a classical MYC target, so why MYC suppresses lactate secretion in these cells is mysterious. Given that the major conclusion of the paper is that hypoxia brings about unexpected metabolic effects in non-transformed cells, I feel there needs to be more clarity about how these effects occur.

We agree that this work generates additional questions related to the mechanisms of hypoxia-mediated MYC signaling and the metabolic consequences of MYC activation in both normoxia and hypoxia. We feel that our present data strongly support the conclusion that MYC uncouples an increase in HIF-dependent glycolytic gene transcription from glycolytic flux in hypoxia. Based on the substantial body of work presented here, we think this finding stands on its own while providing a foundation for additional investigations. The upstream regulation of MYC is complex and involves genetic, transcriptional, translational, and post-translational mechanisms. We have now included a paragraph in the discussion reviewing relevant information but consider further investigation into these mechanisms beyond the scope of the current manuscript. Similarly, given the protean effects of MYC on gene transcription, dissecting the metabolic and transcriptional consequences of its activation in normoxia and hypoxia would require substantial time and resources. We suggest in the manuscript that hypoxia-induced MYC activation couples cell metabolism to proliferation rates. This hypothesis points to fundamental cell biological questions best addressed by future studies.

Reviewer #3 (Recommendations for the authors):I felt the strengths of this paper are substantial, and the impact of the findings will set the stage for further studies contrasting the molecular and metabolic effects of hypoxia in primary vs. transformed cells.1. Since the paper is predicated on the comparison of primary vs. transformed cells, it would be appropriate if the authors could repeat some key experiments with transformed cells to show the side-by-side difference of primary vs. transformed cells in MYC-dependent reduction of glycolytic activity.

We appreciate this suggestion. We have adopted our metabolic flux analysis approach from several studies on cancer cells that have been published previously and cited in the present manuscript. In addition, the roles of HIF and MYC have been defined primarily through work on cancer cell biology. Indeed, the literature on these subjects is vast. Rather than repeat prior experiments, we have attempted to cite the most relevant publications in our manuscript.

2. Is this MYC-dependent reduction of glycolytic activity also similarly seen in primary cells with fewer mitochondria than LFs and PASMCs (such as endothelium)?

We have attempted to study endothelial cells using a similar approach; however, we have found that these cells do not proliferate in hypoxia. This has precluded a detailed metabolic flux analysis as we have performed in LFs and PASMCs, and also suggests different metabolic regulatory mechanisms in these cells. In this manuscript, we have focused on two cell types that proliferate despite hypoxia and have found this is due, in part, to sustained MYC signaling. In future work, we plan to examine similar mechanisms in additional cell types.

3. Can the authors confirm that MYC is up-regulated in hypoxic primary tissue in normal physiologic conditions in vivo?

We agree that a limitation of the current work is a lack of in vivo studies supporting MYC signaling in hypoxia. We and others have generated preliminary data implicating MYC signaling in hypoxia-induced pulmonary vascular remodeling. These data include increased MYC transcript and target gene expression in the chronic hypoxia rat model of pulmonary hypertension. We are currently working to explore the role of MYC in hypoxia-induced vascular remodeling in vivo as the subject of future studies.

4. The presentation of results can be a bit difficult to follow, particularly since the authors often will put in commentary and references of known functions of various metabolites in the actual Results. I suggest that they save the commentary for the Discussion and simply report data in the Results.

We appreciate this feedback. We have made a stylistic decision to include some commentary in the results to provide context for the data we have chosen to highlight while reserving the Discussion for bigger themes. We have attempted to streamline the presentation while including data from the additional experiments requested.